# Adar RNA editing-dependent and -independent effects are required for brain and innate immune functions in *Drosophila*

Patricia Deng[1,6], Anzer Khan [2,3,6], Dionna Jacobson [1], Nagraj Sambrani [2], Leeanne McGurk [4], Xianghua Li [4], Aswathy Jayasree[2], Jan Hejatko[2,3], Galit Shohat-Ophir[5], Mary A. O'Connell[2], Jin Billy Li [1✉] & Liam P. Keegan[2✉]

ADAR RNA editing enzymes are high-affinity dsRNA-binding proteins that deaminate adenosines to inosines in pre-mRNA hairpins and also exert editing-independent effects. We generated a *Drosophila Adar^{E374A}* mutant strain encoding a catalytically inactive Adar with CRISPR/Cas9. We demonstrate that Adar adenosine deamination activity is necessary for normal locomotion and prevents age-dependent neurodegeneration. The catalytically inactive protein, when expressed at a higher than physiological level, can rescue neurodegeneration in *Adar* mutants, suggesting also editing-independent effects. Furthermore, loss of Adar RNA editing activity leads to innate immune induction, indicating that *Drosophila* Adar, despite being the homolog of mammalian ADAR2, also has functions similar to mammalian ADAR1. The innate immune induction in fly *Adar* mutants is suppressed by silencing of Dicer-2, which has a RNA helicase domain similar to MDA5 that senses unedited dsRNAs in mammalian *Adar1* mutants. Our work demonstrates that the single Adar enzyme in *Drosophila* unexpectedly has dual functions.

[1] Department of Genetics, Stanford University, Stanford, CA, USA. [2] Central European Institute of Technology, Masaryk University, Brno, Czech Republic. [3] National Centre for Biomolecular Research, Faculty of Science, Masaryk University, Kamenice 5, 625 00 Brno, Czech Republic. [4] MRC Institute of Genetics and Molecular Medicine, Western General Hospital, Crewe Road, Edinburgh EH4 2XU, UK. [5] The Faculty of Life Sciences and The Multidisciplinary Brain Research Center, Bar Ilan University, Ramat Gan, Israel. [6]These authors contributed equally: Patricia Deng, Anzer Khan. ✉email: jin.billy.li@stanford.edu; Liam.Keegan@ceitec.muni.cz

Adenosine-to-inosine (A-to-I) RNA editing is an important co-transcriptional RNA modification (for reviews[1,2]). In A-to-I RNA editing, the ADAR (adenosine deaminase acting on RNA) enzyme binds to double-stranded RNA (dsRNA) and deaminates specific adenosines to inosines. Inosines are generally read by the cellular machinery as guanosines and RNA editing can change encoded protein sequences, as well as altering RNA structure and RNA splicing. ADAR RNA editing occurs throughout the transcriptome at about fifty specific coding sites in humans but site-specific editing is greatly expanded in some invertebrates such as Drosophila, where it occurs at over a thousand specific coding sites in transcripts. Additional promiscuous RNA editing occurs in long dsRNAs formed by inverted repetitive elements in both mammals and Drosophila; in humans this additional ADAR RNA editing occurs at millions of sites, mostly non-specifically and at low efficiency in Alu dsRNA hairpins embedded in protein-coding transcripts[3–5]. Unlike DNA mutations, RNA editing usually changes only some proportion of the transcripts of particular genes, and editing levels at individual sites can be different in different tissues and developmental stages[5–8], due to different ADAR expression levels and other tissue-specific factors. Thus, RNA editing is a powerful tool that can over-write and enrich the genomically encoded information during the process of gene expression.

ADAR proteins also have editing-independent aspects to their functions[9]; ADAR binding to dsRNA and editing of adenosines within the dsRNA are separable events, and ADARs bind to some dsRNAs that are not efficiently edited[10]. Studies of ADAR1 effects on pri-miRNA processing show that ADAR binding to dsRNA can interfere with binding of other proteins such as Drosha and Dicer[9,11]. ADARs strongly affect processing of many pri-miRNAs even though very few mature miRNAs are themselves edited[12,13]. Editing-independent effects are likely to involve ADAR1 binding to RNAs or interactions with other proteins, such as Dicer[14], and it is important to understand such actions of ADARs.

In vertebrates, the major defects in Adar mutants appear to depend almost entirely on Adar RNA editing activity in vivo although the editing-independent effects are also detected. Humans and mice have two catalytically active ADAR proteins: ADAR1 and ADAR2, in addition to ADAR3, which is catalytically inactive. ADAR1 is essential for innate immune function (for review[15,16]). Mice lacking Adar1 die by embryonic day E12.5 due to the aberrant activation of cytoplasmic, antiviral, innate immune dsRNA sensors, which causes an upregulation of interferon expression and innate immune responses[17–19]. Elimination of MAVS, a mitochondria-associated adaptor protein required for innate immune activation by RIG-I-like receptors (RLRs), antiviral cytoplasmic dsRNA sensors, allows Adar1, Mavs double mutant mice to survive to birth, although they die soon after birth[18,19]. The immune regulatory role of ADAR1 is largely dependent on its RNA editing activity; $Adar1^{E861A}$ mice expressing catalytically inactive Adar1 E861A protein also show upregulation of interferon-stimulated genes (ISG) and embryonic lethality[17]. The $Adar1^{E861A}$ mutant mice die two days later than Adar1 null mutants, by E14.5. Elimination of MDA5 (encoded by Ifih1), one of three antiviral dsRNA sensor helicases, allows the $Adar1^{E861A}$, Ifih1 double mutant to survive to a full life span with no obvious defects[17]. The differences between inactive $Adar1^{E861A}$ mutant and Adar1 null mutant mice suggest that editing-independent effects are also significant, although restoring $Adar1^{E861A}$ by transfection in Adar1 mutant cells prevents aberrant innate immune induction only very partially[18].

In contrast to the innate immune role of Adar1, vertebrate Adar2 is critical for neurological function and shows an even clearer dependence on RNA editing activity. Mice lacking Adar2 die within 3 weeks of birth with increasingly severe seizures

associated with effects of increased influx of calcium ions through glutamate receptors on synaptic plasticity across the brain[20,21]. These severe neurological defects are caused by loss of editing at a single glutamine (Q) to arginine (R) editing site in the Gria2 transcript encoding the key GluA2 subunit of AMPA class glutamate gated ion channels[20].

Drosophila offers an opportunity to study Adar RNA editing-dependent and independent functions in more detail. Unlike mammals, Drosophila has only one Adar gene encoding a protein similar to mammalian ADAR2[1,22,23]. Drosophila Adar null mutants such as $Adar^{5G1}$ have severe locomotion defects, seldom fly, have aberrant neurotransmitter synaptic vesicle accumulation and sleep defects and develop age-dependent neurodegeneration[24,25]. All $Adar^{5G1}$ null mutant phenotypes examined are rescued by expressing Drosophila Adar or human ADAR2 but not human ADAR1[22], showing the strong conservation of function between these ADAR2-type enzymes.

To determine whether Adar mutant phenotypes in Drosophila are fully dependent on RNA editing, we generated a fly strain ($Adar^{E374A}$) that has a point mutation in the chromosomal Adar gene resulting in production of catalytically inactive Adar E374A enzyme. We find that loss of Adar RNA editing activity in Drosophila leads to severe locomotion defects and age-dependent neurodegeneration similar to $Adar^{5G1}$ null mutant flies[24]. Furthermore, we show that the loss of RNA editing by Adar induces aberrant innate immune gene expression in Drosophila, through Dicer-2 acting as a dsRNA innate immune sensor. This is analogous to the findings in mice where loss of Adar1 editing leads to immune induction through the activation of MDA5[17]; Drosophila Dicer-2 has a helicase domain of the same type as MDA5[26]. These results show that Drosophila Adar has dual functions of both Adar1 and Adar2 proteins in vertebrates.

## Results

**An $Adar^{E374A}$ mutant encoding a catalytically inactive Adar.** Drosophila deletion mutants with low to completely depleted levels of Adar have been used for characterizing Adar functions[24,27]. However, it is unclear whether Drosophila Adar functions are dependent on RNA editing activity. To uncouple the editing-dependent from the editing-independent functions of fly Adar, we used CRISPR/Cas9-mediated homologous recombination to generate a fly strain ($Adar^{E374A}$) with a point mutation (Fig. 1a, b), in the catalytic Adar adenosine deaminase domain at the active site glutamate that is critical for transferring a proton during the adenosine deamination reaction. In addition, a synonymous point mutation was introduced at the sgRNA target site to prevent repeated cleavage by Cas9. This Drosophila $Adar^{E374A}$ mutant encodes a stable protein functionally similar to a catalytically inactive $Adar1^{E861A}$ mouse mutant, which has been critical for showing that ADAR1 has both editing-dependent and editing-independent effects[17].

The Adar gene is on the first (or X) chromosome and we confirmed that RNA editing was eliminated by sequencing of total, ribosome-depleted head RNA of male $Adar^{E374A}$ and wild type $w^{1118}$ flies. Wild-type flies have site-specific editing events with a wide range of editing levels in hundreds of transcripts (Fig. 1c), as previously described[28], but $Adar^{E374A}$ flies undergo no detectable A-to-I editing (Fig. 1d). The expression level of the $Adar^{E374A}$ transcript is slightly increased compared to wild type Adar (Supplementary Fig. 1).

**RNA editing is required for CNS function.** For a simple assessment of $Adar^{E374A}$ catalytically inactive and $Adar^{5G1}$ null mutant CNS function, we examined locomotion in male flies. First, as in the original study that reported the $Adar^{5G1}$ null

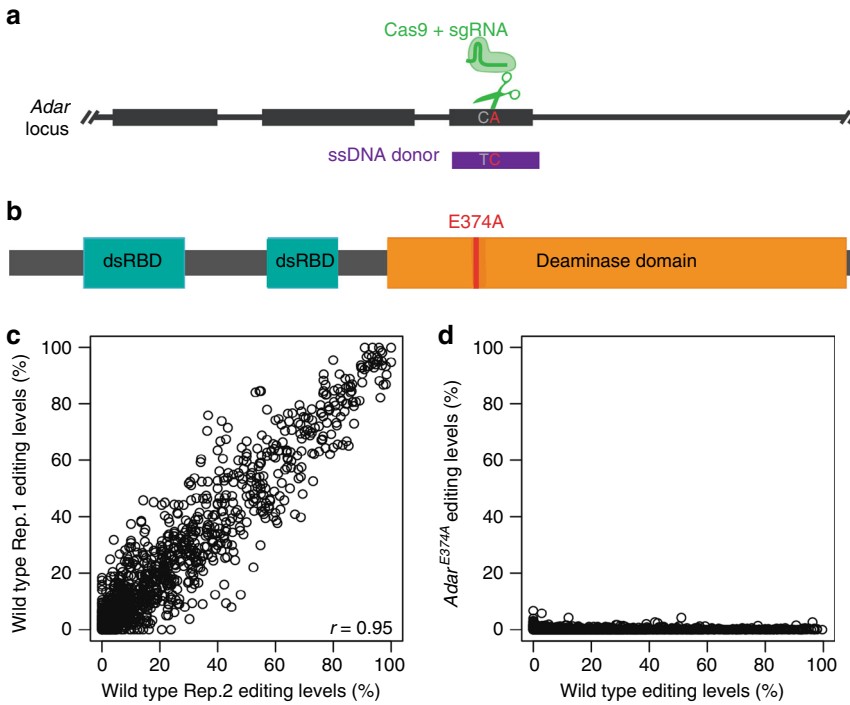

**Fig. 1 A point mutation in the ADAR deaminase domain prevents all RNA editing. a** Schematic of the DNA mutations made with the Cas9/CRISPR system. The *Adar* locus is depicted in black, the sgRNA and Cas9 are green, and the DNA donor is purple. The mutations, shown in gray and red, are C1725T (a synonymous change to the PAM region to prevent re-cutting by the CRISPR system) and A1733C (leading to the E374A mutation). The CRISPR target region is on the reverse strand. **b** Schematic of the domain structure of fly Adar and the E374A mutation made to the deaminase domain. **c** Plot of RNA editing levels in two representative control samples. Pearson's correlation is shown. **d** Plot of RNA editing levels in the wild type versus $Adar^{E374A}$ mutant.

mutant[24], we conducted open field locomotion assays by counting the number of lines crossed by individual flies in a gridded petri plate (Fig. 2a). Each fly was observed for three periods of two minutes each; each period began with tapping the petri plate on the bench but the locomotion is otherwise unstimulated and spontaneous. Neither the $Adar^{E374A}$ nor the $Adar^{5G1}$ mutants moved far compared to controls, which indicates their severe locomotion defects (Fig. 2a).

Next, we conducted negative geotaxis assays, commonly used measures of behavioral health, coordination, and neurodegeneration[29]. These assays take advantage of the instinctive climbing behavior of *Drosophila*, which stimulates them to climb up the walls of vials after they are tapped down. In this assay, flies were placed into glass cylinders, which were tapped on a table, and the number of flies above the 10.5 cm goal height after 30 s was recorded (Fig. 2b). Both $Adar^{E374A}$ and $Adar^{5G1}$ flies could barely climb and almost never reached the goal height in the 30 s time periods, whereas wild type flies usually did (Fig. 2b). In addition, similar to the $Adar^{5G1}$ null mutants[24], $Adar^{E374A}$ mutants undergo heat-sensitive paralysis and they also do not produce progeny. Therefore, our results demonstrate that fly Adar function in the central nervous system is editing-dependent.

**Catalytically inactive Adar impedes rescue by active Adars.** Open field locomotion defects and neurodegeneration in $Adar^{5G1}$ null mutant flies are fully rescued by expression of a *UAS-Adar* construct expressing the main adult wild type *Adar* cDNA isoform under the control of ubiquitously expressed GAL4 drivers[30]. One of the RNA editing events lost in the $Adar^{E374A}$ mutant is at the *Adar S/G* editing site where editing leads to replacement of a serine residue by glycine in the RNA-binding loop on the Adar deaminase domain[23,30–32]. The genome encoded, unedited Adar S isoform is eightfold more active than the Adar G isoform in assays on RNA

editing substrates in vitro[30] and is also less active in comparisons between two fly strains genome-engineered to express the Adar S and Adar G proteins individually[33]. Because the $Adar^{E374A}$ mutant strain produces only the Adar E374A, S isoform, we wished to test the rescue of locomotion in the $Adar^{E374A}$ mutant when expression of catalytically active Adar S is restored by GAL4/UAS-controlled expression from cDNA (Fig. 2a). Expressing the more active, genome-encoded *Adar S* unedited isoform under *actin-5C-Gal4* driver control is lethal with increased editing levels in embryonic and larval transcripts so we instead used the cholinergic neuron-specific *ChAT-Gal4* driver (*choline acetyltransferase promoter-GAL4* previously named *Cha-Gal4*) which drives expression in the numerous brain cholinergic neurons that express the acetylcholine synthesizing enzyme choline acetyltransferase[30]. We note that $Adar^{5G1}$ is close to being fully rescued by *ChAT-Gal4, UAS-Adar 3/4S* (i.e. *ChAT > Adar 3/4S*) expression while $Adar^{E374A}$ is partially rescued (Fig. 2a, b).

The partial rescue of $Adar^{E374A}$ could be due to a dominant negative, interfering effect of having an inactive AdarE374A protein present that may reduce access to editing sites in target RNAs[34,35]. We wished to test the hypothesis that endogenous Adar E374A protein acts as a competitor to the GAL4/UAS-expressed catalytically active Adar protein by using RNAi to prevent expression of the catalytically inactive protein (Fig. 2a). This experiment is possible because $Adar^{5G1}$ null mutant locomotion defects are also rescued by expressing human ADAR2[22], which has a different sequence from *Drosophila Adar*. Therefore, we expressed human ADAR2 under *ChAT-GAL4* driver control in the $Adar^{E374A}$ strain, with or without an additional *UAS-Adar RNAi* transgene driving knockdown of the *Drosophila Adar* transcript and measured open field locomotion (Fig. 2a). It is clear that human ADAR2 rescues $Adar^{E374A}$ mutant locomotion defects better when $Adar^{E374A}$ transcripts are

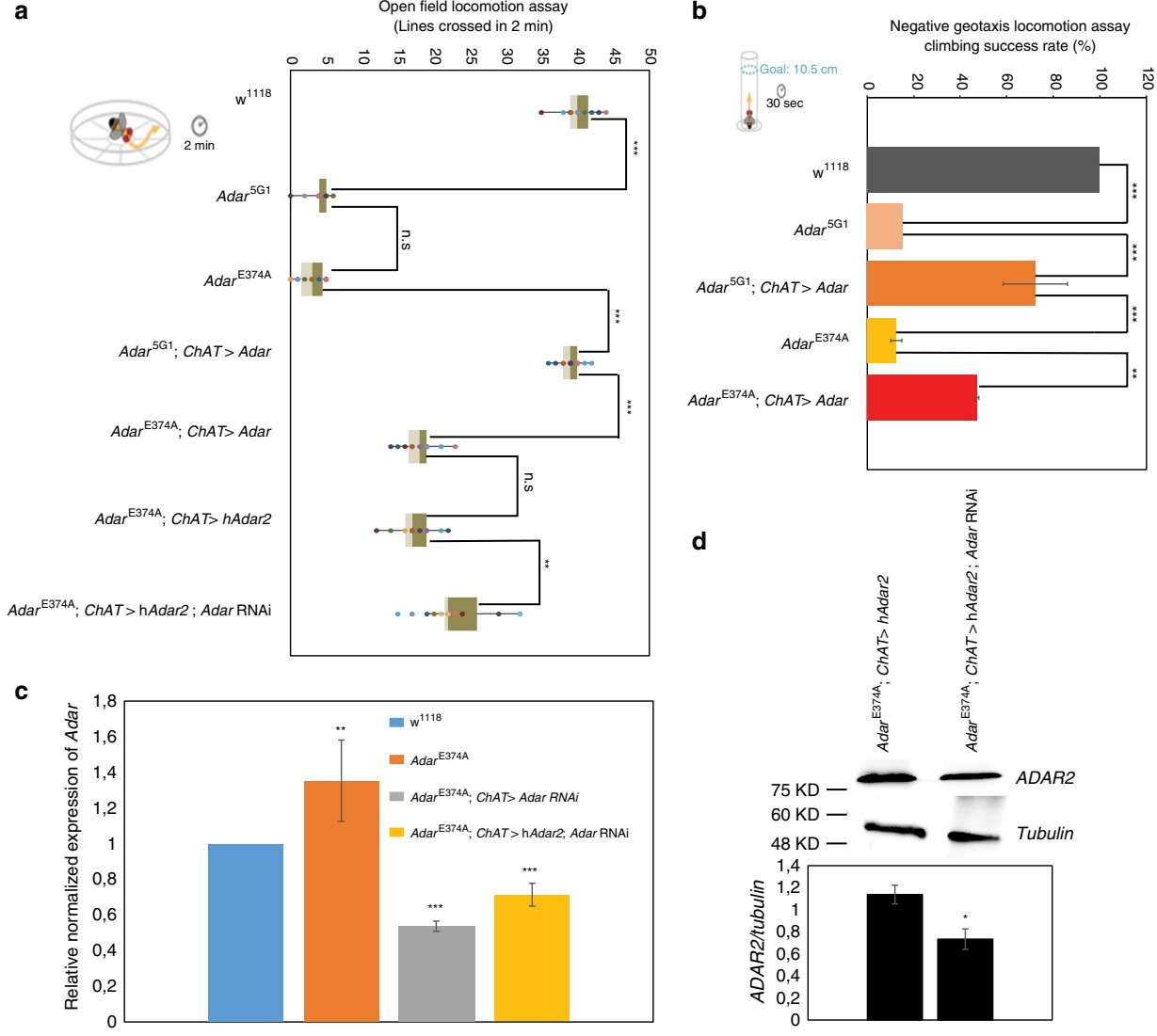

**Fig. 2 Locomotion defects in *Adar*E374A and *Adar*5G1 null mutants. a** Left: Depiction of the open field locomotion assay, in which flies are placed in petri dishes and the distance traveled in 2 min is measured by counting the number of lines crossed. Right: The number of lines crossed by flies in 2 min in the open field locomotion assay; for wild type $w^{1118}$, $Adar^{5G1}$, $Adar^{E374A}$, $Adar^{5G1}$ expressing *Adar 3/4S* under the control of the *ChAT-Gal4* driver (*ChAT > Adar 3/4S*), $Adar^{E374A}$ expressing *Adar 3/4S* under the control of the *ChAT-Gal4* driver, $Adar^{E374A}$ expressing human ADAR2 under the control of the *ChAT-Gal4* without or with an additional *UAS-dAdar* RNAi transgene knocking down dAdar. *p*-values were calculated by a one-way ANOVA followed by Tukey's test. **b** Left: Depiction of the negative geotaxis assay, in which flies are placed in a cylinder, the cylinder is tapped on the table and the number of flies that are above the 10.5 cm mark after 30 s is recorded. Right: The climbing success rate, or percentage of flies that were above the 10.5 cm mark 30 s after the start of the negative geotaxis assay, for the $Adar^{5G1}$ and $Adar^{E374A}$ mutants, and for these mutants expressing *Adar 3/4S* under the control of the *ChAT-Gal4* driver. *p*-values were calculated by a one-way ANOVA followed by Tukey's test (**c**) *Adar* transcript expression in wild type $w^{1118}$, $Adar^{E374A}$, and *Adar* RNAi knockdown flies $Adar^{E374A}$; *ChAT > Adar RNAi* and $Adar^{E374A}$; *ChAT > hADAR2; Adar RNAi* flies that express both *Adar RNAi* and *hADAR2* under *ChAT-GAL4* control. *p*-value was calculated by Student's *t* test. **d** Immunoblot detection in head protein extracts of ADAR2 protein $Adar^{E374A}$, *ChAT > hADAR2* and when an additional UAS transgene construct is present in $Adar^{E374A}$; *ChAT > hADAR2, Adar RNAi*. Quantitation of the immunoblot data is shown below; levels of ADAR2 compared to tubulin in each of the different head protein extracts. *p*-value was calculated by Student's *t* test. For all the above panels **p*-value < 0.05. ***p*-value < 0.01. ****p*-value < 0.005: n.s— not significant. Error bars: SEM (Standard Error of Mean for biological replicates).

reduced by RNAi. The locomotion rescue is still not as strong as in $Adar^{5G1}$; *ChAT > hADAR2*; however, RT-qPCR analysis of *Adar* transcripts shows that the $Adar^{E374A}$ RNAi knockdown is only fifty percent complete (Fig. 2c). In addition, the presence of a second UAS construct to express dsRNA against *Adar* transcript for the knockdown lowers the effectiveness of the GAL4 driver and gives lower ADAR2 protein expression (Fig. 2d). Nevertheless, these results provide clear evidence that inactive ADAR exerts a negative effect on the editing-dependent locomotion function.

**Loss of Adar activity causes age-dependent neurodegeneration.** We aged male $Adar^{E374A}$ mutant and wild type $w^{1118}$ flies for up to 30 days and prepared haematoxylin and eosin stained, five microns thick paraffin wax head sections from ten flies for each group to visualize neurodegeneration. All the $Adar^{E374A}$ mutant head sections show large vacuoles in various regions of the brain and deterioration in the retina that are not observed in aged wild type flies (Fig. 3a–f). In $Adar^{5G1}$ the neurodegeneration develops first in the retina and in the mushroom body calyces and later in the rest of the brain[22]; in $Adar^{E374A}$ the degeneration may be

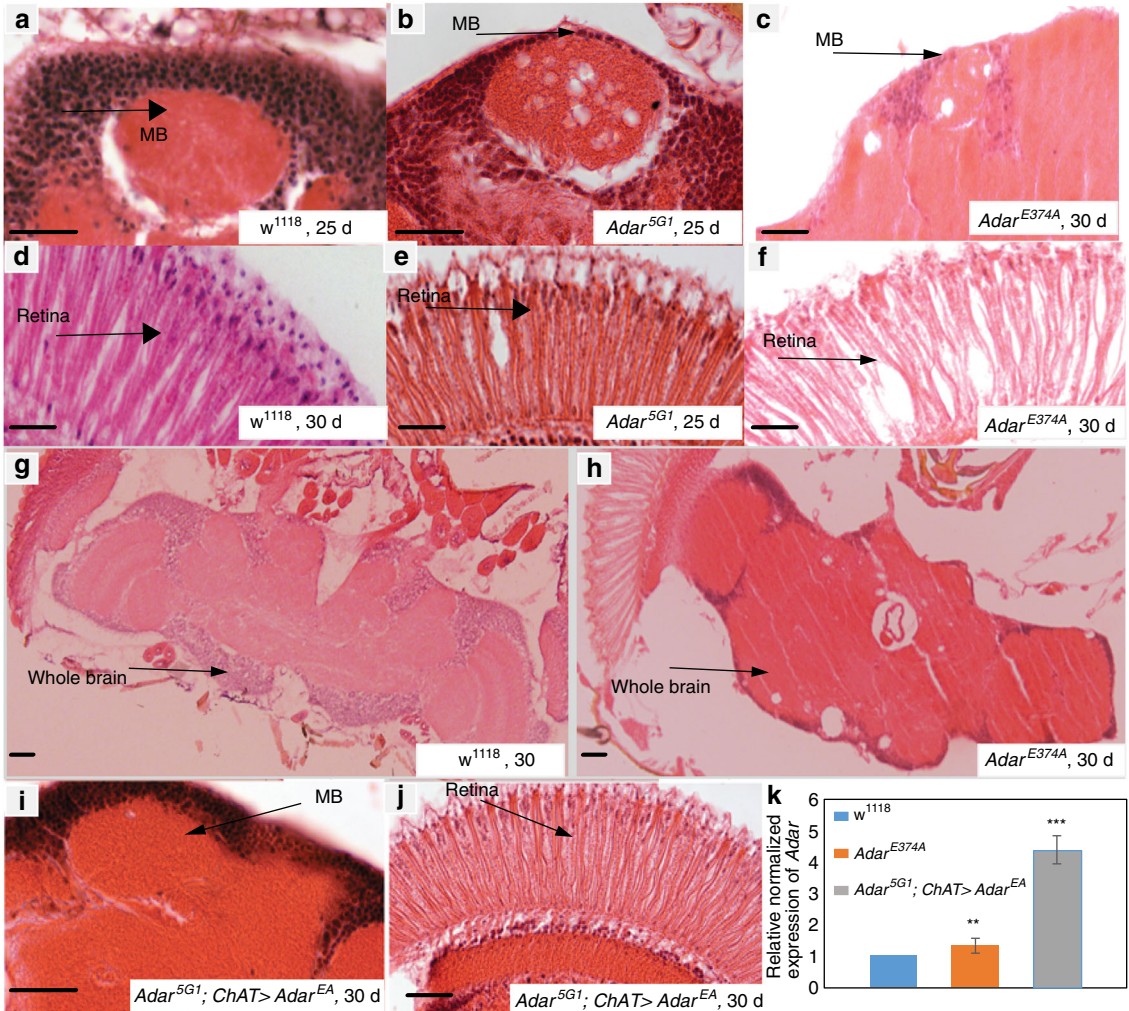

**Fig. 3 Neurodegeneration in the *Adar^E374A* mutant is similar to the *Adar^5G1* null mutant. a–j** Neurodegeneration in haematoxylin-eosin stained 5 micron paraffin wax sections of mushroom body in (**a**) wild-type *w^1118* day 25, (**b**) *Adar^5G1* at day 25 (**c**) *Adar^E374A* at day 30. Staining of retina in (**d**) wild-type *w^1118* day 30 (**e**) *Adar^5G1* at day 25 (**f**) *Adar^E374A* at day 30. Staining of whole brain (**g**) wild-type *w^1118* day 30 (**h**) *Adar^E374A* at day 30. (**i**) *ChAT-GAL4*-driven *Adar^E374A* expression in cholinergic neurons is sufficient to prevent neurodegeneration in *Adar^5G1* mutant mushroom body at day 30 and (**j**) retina day 30. Scale bars: 20 μm. **k** *Adar* transcript expression in fly heads is greater in *Adar^5G1*; *ChAT > Adar^E374A* flies expressing *Adar* from a *UAS-Adar^E374A* construct in cholinergic neurons than it is in wild type *w^1118* or *Adar^E374A* chromosomal mutant flies. *p*-value was calculated by Student's *t* test *: *p*-value < 0.05. **: *p*-value < 0.01. ***: *p*-value < 0.005: n.s—not significant. Error bars: SEM (Standard Error of Mean for biological replicates).

slightly less severe and less focused in the mushroom body calyces. Therefore, neurodegeneration results largely from loss of Adar RNA editing activity when Adar expression levels are near normal, although slight changes in the neurodegeneration in *Adar^E374A* may be due to still having the inactive Adar protein present.

We had previously observed suppression of neurodegeneration in *Adar^5G1*; *ChAT > UAS-Adar^EA* expressing catalytically inactive ADAR from a *UAS-Adar^EA* cDNA transgenic construct under *ChAT-GAL4* driver control (Fig. 3i, j). The *UAS-Adar^EA* transgenic construct expresses a catalytically inactive protein identical to the predominant adult isoform expected to be expressed from the *Adar^E374A* chromosomal mutant. Even though *UAS-Adar^EA* expression is driven specifically in cholinergic neurons, when transcripts are quantitated in whole fly heads, the transgenic *Adar^EA* transcript is 3–4 times as abundant as the *Adar^E374A* transcript in the *Adar^E374A* chromosomal mutant or the *Adar* transcript in wild-type *w^1118*, indicating that *Adar^E367A* transcript expression in cholinergic neurons is above normal levels (Fig. 3k). Evidently, expressing more of the inactive

AdarE367A protein in cholinergic neurons increases the protection against neurodegeneration.

**Aberrant innate immune induction in *Adar* RNA editing mutants.** Next, we hypothesized that, because there are such severe CNS defects in the *Adar* mutants, there might be changes in the expression of neurological genes. As altering a single amino acid in Adar is a less severe change than removing Adar entirely, the CRISPR *Adar^E374A* mutant allows us to focus on editing-dependent changes in gene expression. We performed RNA sequencing of fly heads with a minimum of three biological replicates to examine gene expression differences in the *Adar^E374A* and *Adar^5G1* mutants compared to the wild type *w^1118*. We observed 228 and 751 differentially expressed genes (DEGs) in *Adar^E374A* and *Adar^5G1* respectively (10% FDR, abs(log_2-FoldChange) > = 0.6, Fig. 4a, b, Supplementary Fig. 2). Approximately half (51%) of the 228 genes affected in *Adar^E374A* are also affected in *Adar^5G1*, suggesting that they are affected by loss of RNA editing. Gene expression changes in *Adar^5G1* only are

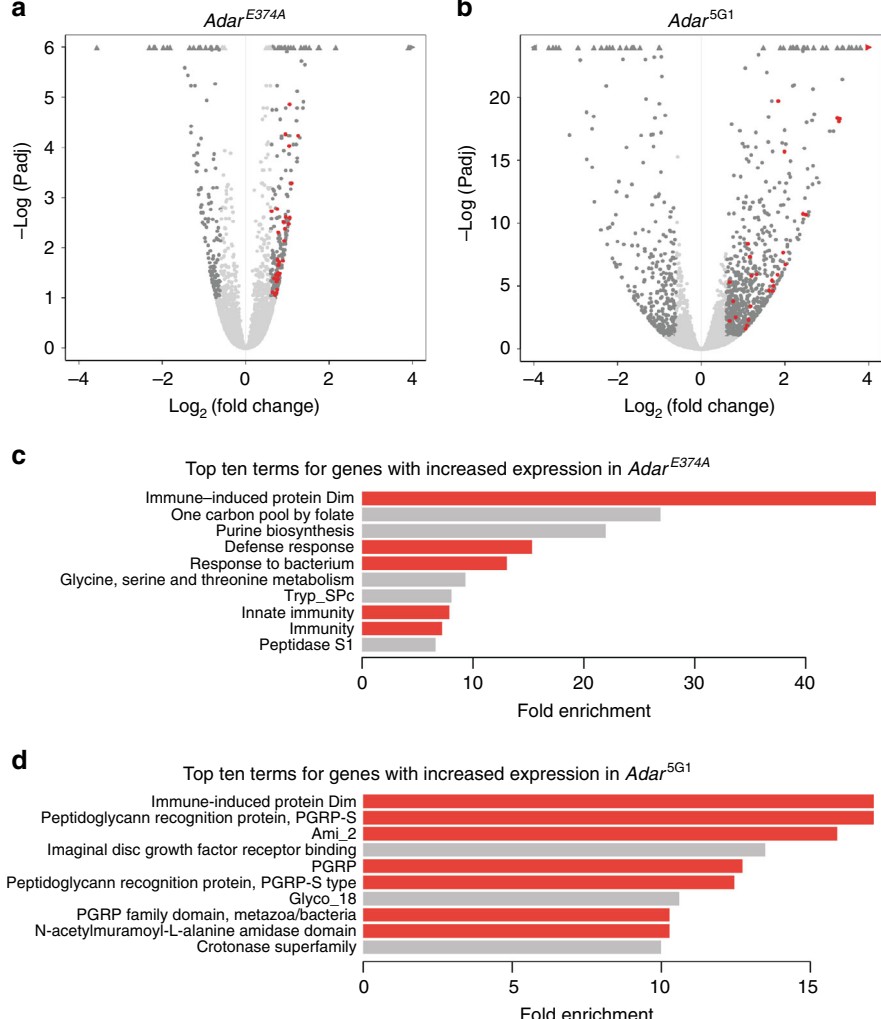

**Fig. 4 Immune gene transcripts in *Adar* mutant heads. a** Volcano plot (-log(adjusted p value) vs. log$_2$(fold change)) for *Adar*[E374A] vs wild type RNAseq. **b** Volcano plot (-log(adjusted *p* value) vs. log$_2$(fold change)) for *Adar*[5G1] vs wild type RNAseq. **a**, **b** Dark gray points indicate genes that are significantly differentially expressed (DESeq2 10% FDR; abs(log$_2$(fold change)) > = 0.6). Red points indicate genes significantly differentially expressed that have immune-related GO terms (highlighted in red in **c**). Triangles indicate points that lie outside the plotted area shown. Less than 0.5% of the points lie outside the plotting area. **c** The top 10 out of 27 significantly enriched terms for genes that are significantly overexpressed in the *Adar*[E374A] mutant. Bars are colored red for immune-related terms. (DESeq2: 10% FDR; DAVID: log$_2$FoldChange > = 0.6, Benjamini adjusted *p*-value < 0.01) The terms used in the analysis are the DAVID default terms (biological process, molecular function, and cellular component GO terms, clusters of orthologous groups (COG), UniProt keywords and sequence features, Kyoto Encyclopedia of Genes and Genomes (KEGG) pathways, InterPro, Protein Information Resource (PIR) superfamily, and SMART). **d** The top 10 out of 84 significantly enriched terms for genes with increased expression in the *Adar*[5G1] mutant. Bars are colored red for immune-related terms. "PGRP" is "Peptidoglycan recognition protein." (DESeq2: 10% FDR; DAVID: log$_2$FoldChange > = 0.6, Benjamini adjusted *p*-value < 0.01) The terms used in the analysis are the same as in (**c**).

likely a response to the loss of Adar protein, but not solely due to loss of RNA editing activity.

We used DAVID Gene Ontology (GO) term analysis to examine the functions and categories of the most differentially expressed genes in the *Adar*[E374A] mutant (Benjamini adjusted *p*-val < 0.01, abs(log$_2$FoldChange) > = 0.6). Although *Adar* mutants display CNS defects and RNA editing sites are enriched in neuronal genes in *Drosophila*[8], we did not observe an enrichment of neurological terms in the genes with increased or decreased expression in the *Adar*[E374A] mutant (Fig. 4c, d, Supplementary Fig. 2b, c). However, we did find an enrichment of neurological terms in genes with differential exon usage, indicating that many neurological genes are differentially spliced in ways that depend on Adar RNA editing activity (Supplementary Fig. 3). Such affects could be due to inosine pairing like guanosine, leading to gain or loss of splice sites or to editing effects on pre-mRNA structure.

Surprisingly, the most enriched term associated with genes with increased expression in *Adar*[E374A] was "Immune-induced protein" (DIM, or *Drosophila* Immune-induced Molecule) (Fig. 4c). Similarly, the DIMs are most enriched among genes with increased expression in *Adar*[5G1] (Fig. 4d). The 9 DIMs that are significantly overexpressed in *Adar*[E374A] are also over-expressed in *Adar*[5G1], further suggesting that their upregulation is dependent on the loss of RNA editing. In fact, 5 of the top 10 most enriched terms for genes overexpressed in *Adar*[E374A], along with 7 of the top 10 terms for *Adar*[5G1], are related to immune functions (Fig. 4c, d), strongly suggesting that loss of Adar RNA editing induces the expression of immune-related genes in *Drosophila*. For the genes with increased expression in both *Adar*[E374A] and *Adar*[5G1] compared to the wild type, the enrichment in immune terms is even more apparent (Supplementary Fig. 2a). For genes with decreased expression in both

$Adar^{E374A}$ and $Adar^{5G1}$ compared to the wild type, we did not observe any enrichment of GO terms with significance.

Our RNA sequencing experiments were performed on head samples to identify gene expression changes particularly in neuronal genes. However, the aberrant DIM expression in heads could still be outside of neurons in other cells such as fat cells or digestive system; if this is the case, there may also be strong aberrant DIM expression in the rest of the body. Therefore, we first conducted qRT-PCR to examine the immune induction in whole flies, examining expression levels of the nine DIMs that had significantly increased expression levels in the $Adar^{E374A}$ head RNA-seq data (Fig. 4a, left). All of the 9 DIMs tested had substantially increased expression levels in the $Adar^{E374A}$ mutants compared to the wild type. Despite the high variation in the degree of immune gene induction, we observed a consistent trend of upregulation of all 9 genes, including five that are statistically significant. In addition, we surveyed the expression of six other innate immune genes that were not significantly differentially expressed in the head RNA-seq data, and found that four of them were significantly upregulated in the whole animal of the $Adar^{E374A}$ mutant (Fig. 5a, right). Next, we examined the expression changes in the $Adar^{5G1}$ mutant for the nine DIMs and six other innate immune genes (Fig. 5b). Overall, we observed the same trend as we did in the $Adar^{E374A}$ mutant, although the level of gene expression changes may differ for some genes.

DIMs are a broad category of immune genes that includes many uncharacterized or little-studied transcripts. The most studied DIMs are the inducible Anti-Microbial Peptides (AMPs) that kill bacteria. Different AMP transcripts are transcriptionally induced by different pathogens through different defined innate immune signaling pathways. We compared the induction of Mtk (Metchnikowin, normally induced by the Toll-Dif pathway in response to infection with Gram positive bacteria or fungi) in RNA from heads versus whole bodies of Adar mutant flies. We observed that the aberrant induction is much stronger in heads (Fig. 5c), indicating that the aberrant innate immune induction may, after all, be mainly in the brain. Aberrant Mtk induction is much stronger in $Adar^{5G1}$ than in $Adar^{E374A}$, indicating that inactive Adar protein may reduce the aberrant innate immune induction. Similar head-enrichment and weaker induction in the presence of inactive Adar protein is observed for Att-C (Attacin-C, normally induced through IMD-Rel in response to infection with Gram negative bacteria or through Jak-Stat signaling) (Fig. 5d). The Vago transcript which is induced specifically by the distinct antiviral signaling pathway activated downstream of the Dicer-2 innate immune sensor[36,37], and not by Toll or Imd signaling, is also induced in Adar mutant head RNA (Fig. 5e). We conclude that a set of AMP innate immune transcripts are aberrantly upregulated in fly heads due to the loss of RNA editing; these include AMPs usually activated by either the IMD-Rel or Toll-Dif pathways as well as Vago which indicates aberrant activation through Dicer-2 acting as an antiviral dsRNA-activated innate immune sensor.

With regard to aberrant AMP transcript induction, we had also previously observed that expression of $UAS-Adar^{EA}$ ubiquitously under armadillo-GAL4 driver control suppresses aberrant expression of AMPs as effectively as UAS-Adar wild type expression (Fig. 5f), although TotC (Turandot-C), which is considered more of a stress-induced transcript is not well suppressed. We conclude that a partial inhibition of aberrant Dcr-2 innate immune signaling by AdarE374A protein when expressed at normal levels, becomes much stronger when the catalytically inactive protein is expressed ubiquitously from UAS constructs under arm-GAL4 driver control. This nearly total suppression of innate immune induction by increased inactive Adar (Fig. 5f) is very striking.

**Dicer-2 acts as an innate immune sensor in Adar mutant flies.** Drosophila does not have a homolog of the vertebrate antiviral cytoplasmic dsRNA sensor Mda5 helicase; the most closely related helicase in Drosophila is Dicer-2, which has been proposed to act as an innate immune sensor for RNAs of some specific viruses[36,38]. To test whether Dicer-2 is involved in the aberrant innate immune induction in Adar mutants we crossed $Adar^{E374A}$ and $Adar^{5G1}$ to the $Dcr-2^{R416X}$ strain. $Dcr-2^{R416X}$ is a point mutation truncating the Dicer-2 protein within the third Hel2 subdomain of the tripartite N-terminal RNA helicase. We found that Adar mutant locomotion defects are not rescued in the Adar; $Dcr-2^{R416X}$ double mutants (Fig. 6a). In fact, AMP transcript expression is increased rather than decreased in these double mutants (Fig. 6b) compared to the corresponding Adar mutant, especially in the $Adar^{5G1}$ null mutant background.

We considered the possibility that the $Dicer2^{R416X}$ mutant produces a truncated protein having the Dcr-2 Hel1 and Hel2i dsRNA-activated helicase subdomains[26,39] that might still function as a dsRNA sensor. This truncated protein would lack most of Dicer-2, including the RNAseIII domains, and would be defective for siRNA processing so that long dsRNA would increase in this mutant. If Dcr-2 (1-416) still has dsRNA sensor activity then the increased long dsRNA could cause the increased innate immune induction in the $Adar^{5G1}$; $Dcr-2^{R416X}$ double mutant. Intriguingly this increased immune induction is much weaker in the $Adar^{E374A}$; $Dcr-2^{R416X}$ double mutant, suggesting that it is prevented by catalytically inactive Adar protein. To test this idea, we knocked down Dcr-2 by RNAi in cholinergic neurons under ChAT-GAL4 driver control; The knockdown of Dicer-2 transcripts in head RNA is 40–60% complete (Supplementary Fig. 4a), but cholinergic neurons are only a proportion of the cells in a fly head so the Dicer-2 knockdown in cholinergic neurons must be much more complete. AMP transcripts are significantly reduced in $Adar^{E374A}$; $ChAT > Dicer2^{RNAi}$ and $Adar^{5G1}$; $ChAT > Dicer2^{RNAi}$ heads (Fig. 6c). We conclude that aberrant innate immune induction in Adar mutants is likely to involve innate immune signaling from Dcr-2.

Signaling downstream of activated Dicer-2 is known to activate expression of Vago which acts as an interferon-like secreted antiviral signaling protein activating the Jak-STAT pathway[37,40]. Vago expression is significantly reduced in $Adar^{E374A}$; $ChAT > Dicer2^{RNAi}$ and $Adar^{5G1}$; $ChAT > Dicer2^{RNAi}$ heads (Supplementary Fig. 4b). The AMPs aberrantly induced in Adar mutants, like those induced during virus infections, include a small subset of those normally induced by bacterial or fungal infections. These include Att-C that is typically activated by JAK-Stat signaling, Dpt (Diptericin) that is usually activated by the IMD-Rel pathway, and Drs (Drosomycin) that is usually activated through the Toll-Dif pathway (Fig. 6c). Aberrant AMP induction in Adar mutants involves the Jak-STAT pathway as $Adar^{E374A}$; $ChAT > Stat93E$ RNAi flies having RNAi knockdown of Stat93E showed reduced aberrant expression of Attacin-C, Metchnikowin and Drosomycin transcripts; parallel RNAi knockdowns of Rel or Dif did not reduce expression of these same AMP transcripts (Supplementary Fig. 5).

## Discussion

We generated and characterized a Drosophila chromosomal $Adar^{E374A}$ mutant line expressing a catalytically inactive Adar enzyme in order to separate the editing-dependent functions of Adar from the editing-independent functions. Drosophila has only one Adar gene and no Adar RNA editing remains in the $Adar^{E374A}$ CRISPR mutant. We show that all aspects of the $Adar^{E374A}$ mutant phenotype are similar to those described in the $Adar^{5G1}$ null mutant. Furthermore, $Adar^{E374A}$ and $Adar^{5G1}$ flies have defects

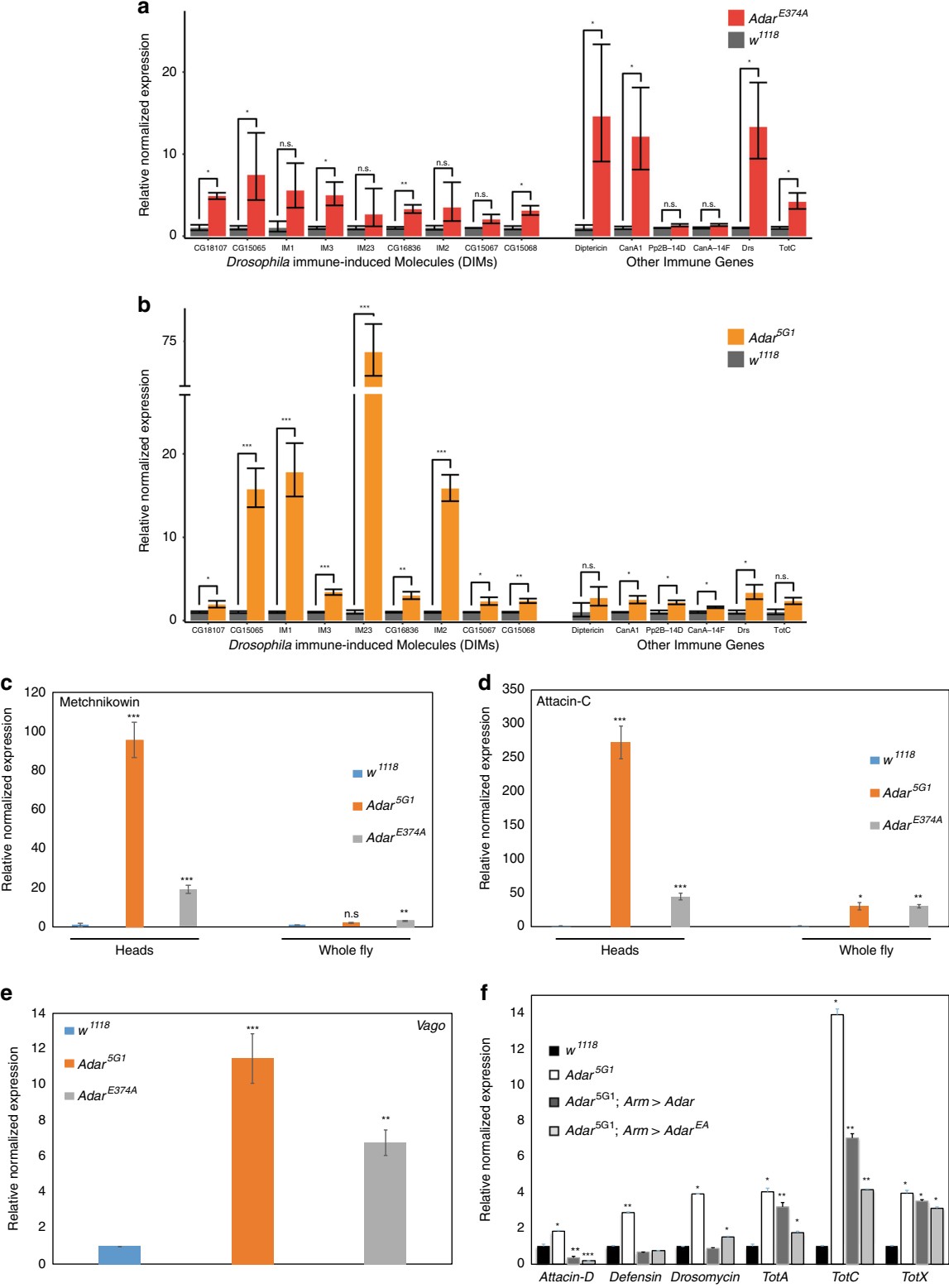

not only in CNS but also in innate immunity, the two systems most affected by *Adar* mutations in vertebrates. In particular, *Adar* mutant flies show Dicer-2-dependent aberrant innate immune induction analogous to that caused by unedited dsRNA activating the innate immune sensor MDA5 in *Adar1* mutant mice. Therefore, the single fly Adar enzyme has dual functions achieved by Adar1 and Adar2 in mammals, and these functions are dependent on RNA editing activity when Adar is expressed normally from the

chromosomal *Adar* locus. Since expressing human *ADAR2* in *Adar*[5G1] mutant flies rescues all fly *Adar* mutant phenotypes tested it seems likely that the involvement of *Drosophila* Adar and not mouse Adar2 in innate immune responses is probably not mainly due to differences between these homologous proteins in vertebrates and flies but due to the particular set of RNAs edited by Adar in *Drosophila*. Association of ADAR1 type proteins with innate immunity and ADAR2 type proteins with neurological

**Fig. 5 Expression of innate immune transcripts in *Adar* mutant flies.** RT-qPCR on RNA from 2-3 days old whole flies. **a** Normalized expression levels for the *Adar*[E374A] mutant (red) relative to wild type (gray) for DIMs that are significantly upregulated in the RNAseq data (left) and other immune genes (right). **b** Normalized expression levels for *Adar*[5G1] (orange) relative to wild type (red) for the same genes as in (**a**). The *y*-axis is discontinuous; the break in the *y*-axis represents the jump in the *y*-axis scale. Below the break, the y-axis scale is the same as in (**a**). *p* value was calculated by Student's *t* test *p-value < 0.05. **p-value < 0.01. ***p-value < 0.005. Error bars: SEM (Standard Error of Mean for biological replicates). **c** *Metchnikowin* transcript expression in *Adar* mutant head or whole body RNA versus in the corresponding wild type RNA sample. **d** *Attacin-C* transcript expression in *Adar* mutant head or whole body RNA versus in the corresponding wild type RNA sample. **e** Aberrant *Vago* transcript expression in *Adar* mutant heads. (**c–e**) levels of each AMP transcript in each of the different whole fly RNA samples is compared to that in *w*[1118]. **f** Expression of either wild type Adar protein or catalytically inactive Adar E374A protein ubiquitously under *armadillo-GAL4* driver expression in *Adar*[5G1] mutant flies strongly suppresses aberrant AMP transcript expression in whole fly RNA. *p*-value was calculated by Student's *t* test *p-value < 0.05. **p-value < 0.01. ***p-value < 0.005: n.s—not significant. Error bars: SEM (Standard Error of Mean for biological replicates).

functions as in vertebrates may have been the original pattern in invertebrates. However, ADAR1 was lost during evolution to insects so ADAR2 proteins may have increased activity on a class of innate immunity-related targets like those edited by ADAR1 in vertebrates and other invertebrates, such as long dsRNAs formed by repetitive sequences.

We observed that although, in general, immune gene induction is seen in both *Adar*[E374A] and *Adar*[5G1] mutants, the degree of induction in whole flies differs between the mutants, suggesting editing-independent Adar effects. For example, the Anti-Microbial Peptide (AMP) transcript inductions are clearly stronger in *Adar*[5G1] than in *Adar*[E374A], suggesting that the presence of catalytically inactive Adar protein may impede the immune induction. The Adar editing-independent effects are most clearly seen when the catalytically inactive Adar[EA] is overexpressed and is able to rescue some *Adar* mutant phenotypes. For example, neurodegeneration (Fig. 3i, j) and aberrant innate immune induction (Fig. 5f) in the *Adar*[5G1] mutant strain are rescued by overexpressing *UAS-Adar*[EA] cDNA constructs using GAL4 drivers. However, this type of rescue experiment involves *Adar* transgene expression at greater than physiological levels. Nevertheless, expressing *UAS-Adar*[EA] cDNA constructs using brain GAL4 drivers did not rescue locomotion defects, consistent with the idea that edited isoforms of CNS proteins are absolutely required for Adar-mediated rescue of locomotion defects. Future work is needed to identify the key editing events that are required for normal locomotion[25,28,41].

Our gene expression analysis reveals an unexpected function of fly Adar, which is to suppress innate immune induction. Although the only fly Adar is most similar to mammalian ADAR2 in terms of both sequence identity and neurological phenotypes, it appears to have an innate immune role in flies similar to that of ADAR1 in vertebrates. *Adar*[5G1] aberrant innate immune induction is likely to result from accumulated unedited intracellular dsRNA in *Adar*[5G1], paralleling the mouse *Adar1* mutant interferon induction through antiviral dsRNA sensors[18], and the human *ADAR1* mutant virus infection mimic syndrome Aicardi Goutières Syndrome[42] and *ADAR1* mutant bilateral striatal neurodegeneration[43]. In our initial attempt to test the role of Dicer-2 in *Adar* mutant innate immune induction the *Dcr-2*[R416X] mutation did not suppress AMP induction in the *Adar*; *Dcr-2*[R416X] double mutants (Fig. 6b). However, our finding that AMP expression was higher in *Adar*; *Dcr-2*[R416X] double mutants than in the corresponding single *Adar* mutant led us to test the effects of *Dicer-2* RNAi knockdown instead. The *Dicer-2* RNAi knockdown experiments showed that the immune induction in the fly *Adar* mutants is mediated by Dicer-2 (Fig. 6c), which has a helicase domain closely related to that of MDA5; these helicases may act as ATP-activated translocases on dsRNA[26] (Fig. 6c). Similar to the vertebrate antiviral dsRNA sensors which are inhibited by dsRNA containing inosine[18,44], ADAR editing of dsRNA inhibits Dicer-2 cleavage in vitro[45], and

may also inhibit innate immune signaling from Dicer-2. A-to-I RNA editing can introduce inosine-uracil wobble base pairs, which weaken dsRNA strand pairing; MDA5-like sensors dissociate more rapidly from imperfectly-paired dsRNAs[46–48]. It is surprising that the Dicer-2 (1-416) protein still mediates innate immune induction but it retains the Hel1 and Hel2i subdomains of the helicase domain and their potential dsRNA contacts; and it may also have interactions with other proteins that help it recognize dsRNA. The Hel2i subdomain of helicase DICER proteins is a hotbed of interactions with dsRBD proteins in, for example, human DICER interacting with specialized dsRBDs of TRBP and PACT and with dsRBD3 of human ADAR1, which affects microRNA and siRNA processing[14]. When expressed in *Drosophila*, human ADAR1 strongly inhibits Dicer-2-mediated RNA silencing through siRNA production, even when catalytically inactive[9] and it may also inhibit innate immune signaling by Dcr-2. We show here that *Drosophila* Adar[EA] also strongly blocks Dcr-2 innate immune signaling (Fig. 5f). Failure to rescue *Adar* mutant aberrant innate immune induction in the *Adar*; *Dicer2*[416×] double mutant is surprising because this *Dcr-2* mutant has been reported to suppress aberrant innate immune induction by long dsRNA overexpressed in *Drosophila*[49]. We do not know the reason for this difference; it may be due to different dsRNAs driving the immune induction or to different levels of dsRNA expression.

An additional way whereby innate immune induction could contribute to all *Adar*[5G1] mutant phenotypes is described in a related manuscript[50]. That manuscript shows that reduced autophagy in *Adar*[5G1] leads to synaptic defects with aberrant accumulation of neurotransmitter synaptic vesicles and associated proteins. In mammalian cells innate immune induction impedes canonical autophagy by diverting the p62 receptor for ubiquitinated proteins to instead form a cytoplasmic innate immune signaling platform[51]; this cross-regulation of p62 by innate immune signaling helps to redirect autophagy to innate immune defense when required. Dicer-2-mediated aberrant innate immune induction in *Drosophila* may also impair autophagy in flies. Determining to what extent all the *Adar* mutant phenotypes arise from aberrant AMP expression leading to neuronal damage or from antiviral innate immune impedance of autophagy, leading to synaptic defects in neurons, requires *Adar*[5G1]; *Dicer-2*[null] double mutants to block all immune induction; however, the work described here shows that no such *Dicer-2* null allele has been characterized yet.

Given the evolutionary expansion of site-specific RNA editing events in *Drosophila*, it one would presume that *Adar* mutant phenotypes would be entirely due to loss of RNA editing in CNS transcripts and to loss of the edited isoforms of encoded proteins. Therefore, it comes as a surprise to find that an important aspect of the *Adar* mutant phenotype is still an aberrant innate immune induction by unedited dsRNA. This surprise also adds renewed interest to the question of how much of the *Adar* null mutant phenotype is solely due to loss of edited isoforms of so many CNS proteins. It was already surprising that *Adar* null mutants flies

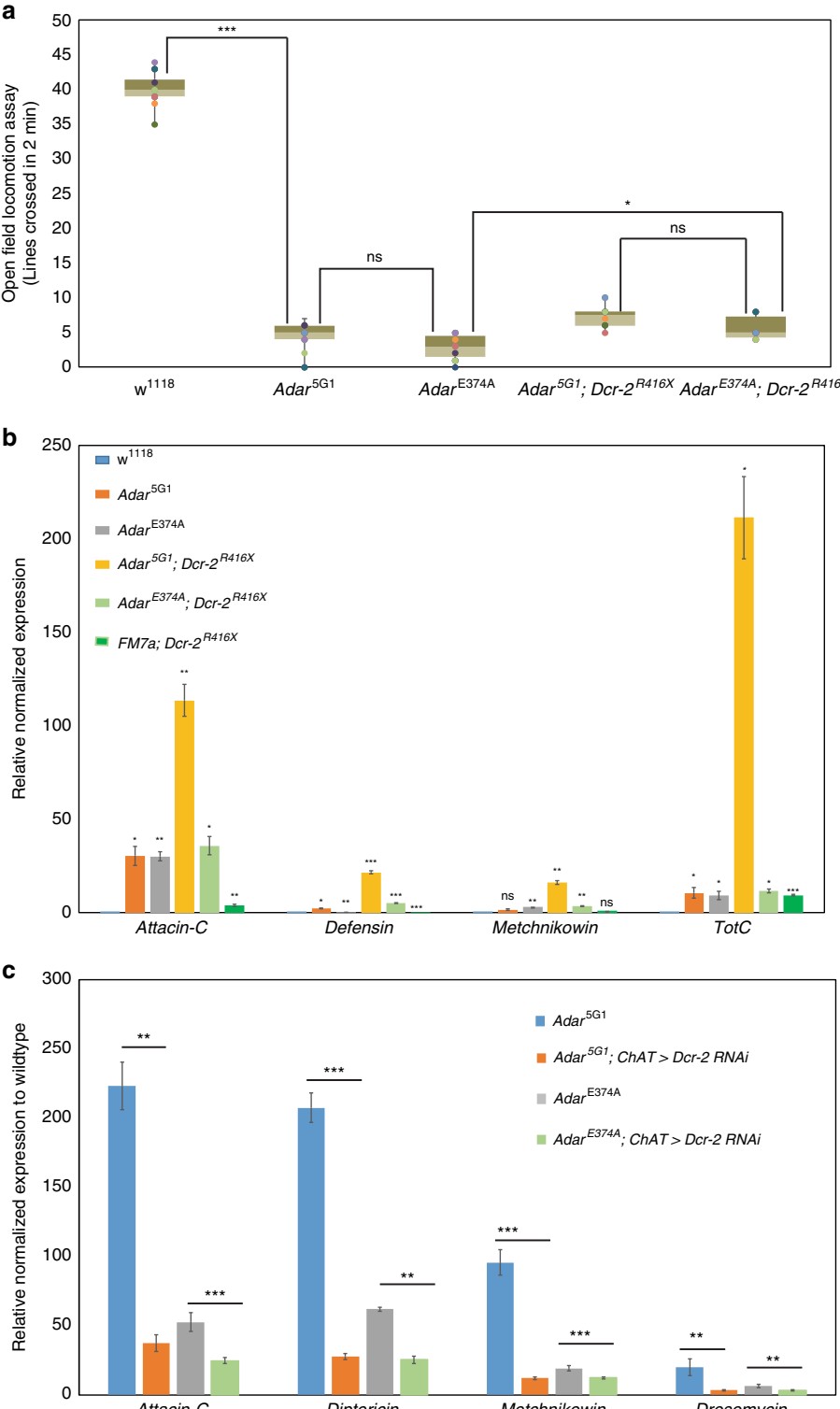

**Fig. 6 *Adar* mutant aberrant AMP expression requires Dcr-2. a** Open field locomotion of *Adar* mutant and *Adar; Dicer2* double mutant flies. *p*-values were calculated by a one-way ANOVA followed by Tukey's test. *p-value < 0.05. **p-value < 0.01. ***p-value < 0.005: n.s—not significant. **b** DIM transcript expression RT-qPCR measurements in *Adar* mutant and *Adar; Dicer-2* double mutant flies. qPCR from 2 day old whole flies. Levels of each AMP transcript in each of the different whole fly RNA samples is compared to that in *w1118*. **c** Rescue of aberrant AMP transcript expression by *Dcr-2 RNAi* knockdown in *Adar*5G1 heads. Levels of each AMP transcript in each of the different fly head RNA samples is compared to that in *w1118*. Decrease in the aberrant induction of AMP transcripts in Adar*E374A*; *ChAT > Dicer2RNAi* and *Adar*5G1; *ChAT > Dicer2RNAi* heads compared to *Adar*E374A and *Adar*5G1, respectively, is statistically significant. *p* value was calculated by Student 's *t* test *p-value < 0.05. **p-value < 0.01. ***p-value < 0.005: n.s—not significant. Error bars: SEM (Standard Error of Mean for biological replicates).

are viable, suggesting that the RNA editing events have been evolutionarily selected so that basic neuronal functions do not depend on the edited protein isoforms. The new findings now suggest that edited protein isoforms fulfill even more specific roles[25,52,53] that will need to be defined within specific neuronal circuits.

## Methods

**Fly stocks and sample collection**. All stocks were grown on standard fly media. *Adar*[5G1] ("*Adar* null") flies are from[31]. The *Cha-Gal4* driver expresses GAL4 in cholinergic neurons under the control of the *choline acetyl transferase* promoter (*Cha* is now named *ChAT* in Flybase http://flybase.org/reports/FBti0024050.html); this GAL4 driver line is Bloomington Stock number 6798, w1118; P[21,48]s19B/CyO, P{sevRas1.V12}FK1. The *UAS-Adar 3/4* transgene expresses an editable cDNA encoding the adult ADAR 3/4 splice form[30]. The UAS-hADAR2 transgene expresses cDNA encoding human ADAR2. RNAi lines are *Adar* (VDRC-7763), *Dcr-2* (VDRC-25090), *Rel* (VDRC-105491), *Dif* (Bloomington-29514), *Stat92E* (Bloomington-35600). For the innate immune qPCR, flies were grown at 25 °C and collected at 2-3 days old. Total RNA was extracted from 7–15 whole flies or 20 heads using RNAdvance magnetic beads (Agencourt), and this was treated using TURBO DNase (Thermo Fisher Scientific).

For the locomotion assays (negative geotaxis and open field locomotion), flies were kept in 12 h/12 h light/dark cycles at 25 °C. Replicate experiments were performed at the same time of day. Flies were not exposed to carbon dioxide for at least 24 h before the assays.

**Adar[E374A] mutant generation**. The *Adar[E374A]* fly line[28] was generated with plasmid and *nos-Cas9* fly line CRISPR reagents as described in[54]. sgRNA target sequence CCTGGAGACTATTTCAGCAT was inserted into the U6b-sgRNA-short plasmid and injected into the *nos-Cas9* flies (attP2 insertion line) by BestGene. The sgRNA plasmid was injected at 75 ng/ul, and the donor oligo (a ssODN with 60 base homology arms on each side of the mutations C1725T and A1733C, depicted in Fig. 1a) was injected at 750 ng/ul in water. The flies were then crossed multiple times to an FM7a balancer stock (Bloomington #785) to isolate the mutants and then backcrossed for 7 generations to *white Canton S* (Shohat-Ophir lab stock).

**Negative geotaxis assay**. Three to four-day-old male flies were placed in two glass cylinders with a fly vial plug at the bottom. The following was repeated five times for each set of flies: The cylinders were banged onto a mouse pad and after 30 s, the number of flies above the 10.5 cm mark was counted. Then there was a 30 s rest period. The average of the percentage of flies above the 10.5 cm mark over the five trials was the "climbing success rate". 14–38 flies were examined for each genotype.

**Open field locomotion assay**. Two-day-old individual male flies were placed in 35 mm petri dishes divided into eight sections by radii, with an additional center circle. The number of lines crossed in 2 min was recorded three times for each fly, and these values were averaged per fly. 15 flies were examined for each genotype.

**Fly ageing and *Drosophila* head sections and staining**. Flies were aged and then fixed in Carnoy's fixation solution (30% chloroform; 10% acetic acid; 60% absolute ethanol) for 3.5 h and flies were then placed in 95% ethanol for 30 min twice and then they were placed in 100% ethanol for 45 min. Flies were kept in methyl benzoate overnight. On the next day, flies were kept in solution of methyl benzoate and paraffin both in equal volume at 60 °C for 1 h, then transferred to fresh liquid paraffin for 1 h twice at 60 °C and then kept in liquid paraffin overnight at 60 °C. Flies were transferred to a plastic petri dish with fresh liquid paraffin at 60 °C and then allowed to cool down. Paraffin blocks were made from this. Sections of 5-μm thickness were taken using a LEICA RM 2235 rotary microtome and then dried overnight on the slide warmer at 37 °C and then processed for hematoxylin and eosin staining.

**Gene expression and splicing analyses**. Reads and mapped reads are as in[28] (GEO GSE86056). DESeq2[55] were used to obtain gene expression levels. DEXSeq[56,57] was used to examine differential exon usage. To determine the editing event, we required the editing level to be ≥2% in at least one of the datasets. 89% of the editing sites have significantly higher editing levels with this cut off than the typical A-to-G sequencing error rate. Editing levels were only used if there were at least 20 reads at the editing site.

**GO term analysis**. DAVID was used for the GO term analysis. The lists of genes tested by DESeq2 (the genes for which the adjusted *p*-value was not NA) were used as the background. The GO terms used are from these lists (DAVID default): COG_ONTOLOGY, UP_KEYWORDS, UP_SEQ_FEATURE, GOTERM_BP_DIRECT, GOTERM_CC_DIRECT, GOTERM_MF_DIRECT, KEGG_PATHWAY, INTERPRO, PIR_SUPERFAMILY, SMART. *P*-values were corrected for multiple hypothesis testing using the Benjamini method, and only the top statistically significant GO terms (adjusted *p* < 0.01) are shown.

In Fig. 3 and Supplementary Fig. 2, InterPro, Biological Process GO terms, UniProt Biological Process Keywords, and SMART terms were considered immune-related and colored red if they were clustered with "immune response" (GO:0006955), "innate immune response" (GO:0045087), or "immune-induced protein Dim" (InterPro 013172). "Biosynthesis of antibiotics" (KEGG dme01130) was also colored red.

**qPCR**. cDNA was generated with iScript Advanced (Bio-Rad) or RevertAid First Strand cDNA Synthesis Kit (Thermo Scientific). qPCR reactions using the KAPA-SYBR FAST qPCR kit (Kapa Biosystems) or LightCycler® 480 SYBR Green I Master mix and the primers listed in Supplementary Table 1 were used to measure expression levels on a Bio-Rad CFX96 Touch Real-Time PCR machine or The LightCycler® 480. All primers were tested to ensure the efficiency levels. Expression levels were normalized to those of RP49 and t-tests were used for statistical analysis.

**Immunoblotting**. Heads of male flies (minimum 20 flies) of the desired genotype were collected and aged for 2 days and then homogenized in NB Buffer (150 mM NaCl, 50 mM Tris-HCl pH 7.5, 2 mM EDTA, 0.1% NP-40). Protein concentration was determined with Pierce BCA Protein Assay Kit. An equal amount of protein was loaded in each lane of a Tris-Glycine SDS Gel and transferred to a nitrocellulose membrane. The membrane was blocked with 5% Milk, incubated with primary antibody overnight. The next day it the membrane was incubated with secondary antibody and developed with Clarity™ Western Blotting Substrate from Bio-Rad. Anti-ADAR2 (HPA018277) (1:250) anti-Tubulin (Developmental Studies Hybridoma Bank) (1:5000). Imaging was performed with ChemiDoc™ XRS + System and signal intensity was quantified with Image J software and Statistical analyses were done using the *t*-test.

**Statistical analysis**. qPCR data and immunoblot data were analyzed by Student's *t* test. A *p*-value of <0.05 was considered statistically significant. Locomotion and Climbing data *p*-values were calculated by a one-way ANOVA followed by Tukey's test. The significance of differences between variables was described based on *p*-values *$p$-value < 0.05. **$p$-value < 0.005. ***$p$-value < 0.001 and n.s (not significant). Error bars: SEM (Standard Error of Mean for biological replicates).

**Reporting summary**. Further information on research design is available in the Nature Research Reporting Summary linked to this article.

## Data availability

The data that supports the findings of this study are available from the authors upon request. The source data underlying Figs. 2a–d, 3k, 5a–e, 6a and Supplementary Figures S1a, S4 and S5 are provided as a Source Data file. The RNA sequencing data are deposited at the NCBI's GEO under GSE86054.

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

## Acknowledgements

The authors thank Jianquan Ni and the Tsinghua Fly Center for providing fly CRISPR reagents. We thank members of the RNA and Immunity and Li lab, Andrew Fire, Joseph Lipsick, Hunter Fraser, and Huaying Fang for helpful discussions and Anne Sapiro for assistance in figure preparation. This work was supported by the National Institutes of Health, (Grant Numbers R01GM102484, R01GM124215, and R01MH115080), and by the Ellison Medical Foundation, grant number AG-NS-0959-12 (to J.B.L.). P.D. was supported by the Stanford Center for Computational, Evolutionary and Human Genomics (CEHG), National Science Foundation Graduate Research Fellowship (no. DGE-114747), Stanford Genome Training Program (NIH T32 HG000044), and Cell and Molecular Biology Training Program (NIH T32 GM007276). This work was supported by an MRC Capacity Building Area Research Studentship to L. McG; by MND Scotland Prize Studentship award (to LK for XL), and by the Medical Research Council UK (U.1275.01.005.00001.01) to MAO. This work was also supported by the European Union's Seventh Framework Programme for research, technological development and demonstration under grant agreement No 621368 (to MO'C). The work was supported from European Regional Development Fund-Project "National infrastructure for biological and medical imaging" (No. CZ.02.1.01/0.0/0.0/16_013/0001775) and by Czech Science Foundation, project No. 19-16963S to LK.

## Author contributions

P.D., A.K., N.S., D.J., X.L., and L.P.K. performed the experiments and analyzed the data. A.J helped with the brain sections. J.B.L., L.P.K., and M.A.O. supervised the study. P.D., A.K., L.P.K., M.A.O., G.S.-O., and J.B.L. wrote the paper, with input from the other authors.

## Competing interests

The authors declare no competing interests.
