## [Peer Review File · Nature Communications]

Reviewers' comments:

Reviewer #1 (Remarks to the Author):

This study focuses on ADAR RNA editing in *Drosophila melanogaster*. While flies null for the single *Drosophila* ADAR (dADAR) have been extensively characterized, at least according to this paper, flies expressing a genomic version of a catalytically dead dADAR have not been tested. Crispr technology has allowed the authors to isolate such a mutant, and for the most part the data indicate that RNA editing is the key function of dADAR, as measured by careful behavioral assays and visualization of head sections.

The authors also perform RNAseq experiments to determine genes that are misexpressed in the null and catalytically-dead mutants, and find, interestingly, that a large number of innate immune-related genes are misexpressed. This is interesting because in other organisms ADARs are known to keep cellular dsRNA from triggering an immune response. In mammals the ADAR-related aberrant innate immune response can be rescued by knocking out downstream effectors such as MDA5 or MAVS, or in *C. elegans*, DRH-1. To perform similar experiments in flies, the authors attempt to rescue the misexpression of innate immunity genes with a mutation in *Dicer-2*, the *Dicer* known to have antiviral functions in *Drosophila*.

For the most part the experiments are well-performed and represent a significant advance to the field. That said, results with the catalytically dead ADAR mutant are not so surprising, and the most interesting results are those showing aberrant expression of innate immune genes. As detailed below, the quality of the paper would increase if the authors expanded the latter section, at the least providing additional discussion.

Specific comments:

1. The authors find that the behavioral defects in the catalytically dead *AdarE374A* mutant are only partially rescued by *Adar* $\frac{3}{4}$ S, and propose that this may be due to a dominant-negative effect of the catalytic mutant. To enable an RNAi knockdown that is specific for the catalytically dead dADAR2, the authors rescue with human ADAR2, and then knockdown the d*AdarE374A* sequence. A very slight rescue in the locomotion assay (Fig. 2A) is observed. Because the rescue is very slight, one wonders if this rescue is significant. The authors should report some measure of statistical significance for Fig. 2A (I think the p-values that appear at the end of the legend refer to panel B, but regardless, the relevant one is missing). As they authors note, the small increase could also be explained by the fact that the RNAi knockdown is only partial. To lend confidence in the significance of the rescue the authors could directly evaluate the change in mRNA levels of *AdarE374A*. Unless confidence in the observation is increased, the authors should not say that the data "... gives clear evidence...".

2. The observation that innate immune genes are misexpressed in dADAR mutants is the most interesting part of the paper, and the validation of these expression changes using qRT-PCR in whole flies is very nice. That said, the quality of the paper would increase if the authors could expand this part of the paper by adding additional experiments that might increase mechanistic insight, or at the least,

additional discussion. I am particularly focused on results with the Dicer-2 mutant. As the authors certainly know, *Drosophila* Dicer-2 processes endogenous siRNAs as well as viral siRNAs, and the requirements for each are slightly different. Is the mutant the authors used known to affect a Dicer2-mediated antiviral response? For example, does this mutant lead to a decrease in viral siRNAs or viral load? If the Dicer2(R416X) mutant only affected production of endogenous siRNAs, it could actually lead to a greater immune response (as the authors observe) since it would free up Dicer-2 to act on viral dsRNA.

In many respects the connection of *Drosophila* ADAR to the immune response may be more similar to other invertebrates than mammals. Indeed, in *C. elegans* the loss of ADARs increases the number of siRNAs (even in the absence of viral infection), and the resulting increase in silencing can be rescued with a second mutation in RNAi factors that act with Dicer in the antiviral response. Is it known that loss of dADAR increases levels of siRNAs and that this can be rescued by the Dicer2R416X? Finally, it would be helpful if the authors provided a bit more discussion of the particular innate-immune genes observed to have altered expression in the dADAR mutants. Was the *vago* gene that has previously been connected to altered immune response in *Drosophila* misexpressed?

Minor comments:

1. On line 224 I think the reference should be to Fig. 5A, left.
2. On line 241 I think the Dicer2 allele should be 416X
3. The fonts in Figure 2 are very hard to read.

Reviewer #2 (Remarks to the Author):

ADAR RNA editing enzyme binds double stranded RNA for deaminating adenosines to inosine. Inosine is decoded as guanosine which can alter the amino acid sequence of the encoded protein, but in double stranded RNA inosine also disrupts base-pairing which tunes down the immune response against double stranded RNA in mice.

Here, Deng et al now determine that ADAR in *Drosophila* requires its editing capacity to exert its function as mutants for catalytically inactive ADAR show neurodegenerative and locomotion phenotypes similar to ADAR null mutants. When this catalytically inactive ADAR protein is overexpressed, however, the ADAR RNA binding function can suppress these phenotypes indicating editing independent functions of ADAR.

In addition, immune genes are induced in ADAR catalytic dead and null mutants. This phenotype is reminiscent of the induction of immune genes in mice ADAR1 mutants, where the double-stranded RNA sensor MDA5 is required to turn on immune genes. The closest homolog of MDA5 in flies, DICER2, however, is not responsible for turning on immune genes. Hence, flies use a different pathway to turn on immune genes and this function relies on ADAR editing.

The work by Deng et al addresses the important question of whether ADAR exerts functions by binding double-stranded RNA that are independent of RNA editing. To address this question they use a

Drosophila model, which has only one ADAR, unlike mice, which have three. These experiments are well performed and they obtained a clear cut answer that editing is required for ADAR function. These data nicely complement previous work done in the more complex situation of mice with having two active ADARs and highlight the importance of RNA editing.

Importantly, the work of Deng et al further discovers an evolutionary conserved function of ADAR in the regulation of immune genes. Immune genes are also turned on in Drosophila, but unlike in vertebrates this seems not to be mediated by DICER2, the sensor of double-stranded RNA in flies.

Minor issues

P1, line 61: should mention that lethality can be rescued by the edited cDNA

Fig 6B: In the double mutant of Adar null, DICER2, the tested immune genes are dramatically up-regulated, but this induction is not present in the ADAR E374A DICER2 mutant. Does the RNA binding capacity of ADAR E374A prevent this induction? To answer this question the DICER2 mutant control needs to be added. Also, a reference for the allele used is missing.

The discussion contains an number of results. These should be moved to the results section.

Reviewer #3 (Remarks to the Author):

ADAR null model organisms have uncovered many important functions of ADAR. However, it was not clear whether these functions of ADAR were editing dependent or independent. Previously, a transgenic mouse defective in ADAR editing (ADAR-E861A) was shown to be embryonic lethal, demonstrating that the ADAR editing activity plays an essential role in mammals (Liddicoat et al. Science 2015). To examine the function of ADAR editing in flies, Deng et al. generated a novel drosophila strain carrying a point mutation in the catalytic domain of ADAR (ADAR-E374A). With ADAR-E374A flies, Deng et al. demonstrated that ADAR editing is entirely responsible for preventing locomotion defects in flies and mostly (but not entirely) responsible for preventing neurodegeneration. Furthermore, they made an unexpected finding that ADAR editing plays a critical role in suppressing immune responses in flies. This is surprising since drosophila ADAR is similar to mammalian ADAR2 whose known roles lie in the brain, and is less similar to mammalian ADAR1, which plays a role in suppressing innate immune responses. Taken together, this study reveals that in drosophila, ADAR's RNA editing activity is required for proper brain and immune function. The ADAR-E374A drosophila model system is novel, and we believe that the authors' findings are interesting. However, mechanisms underlying the locomotion defect or induction of immunity in ADAR deficient flies should be further explored in depth. Furthermore, the quality of data (quantifying images, confirming protein expression, confirming successful knock-down with siRNA etc.) in some figures need to be significantly improved. Please see our comments below.

Major comments:

1) Similar to how the ADAR2^{-/-} mice defects were rescued by introducing the edited version of Gria2 (Higuchi et al. Nature 2000), could the locomotion defect in ADAR-E374A flies be rescued (at least partially if not entirely) by expressing an edited gene? Are there any edited neuronal genes that are strong candidates? Alternatively, since the authors mention that differential exon usage was most significant among neuronal genes (Supplementary Figure 3), could expressing a properly spliced transcript rescue the locomotion defect in ADAR-E374A flies? We acknowledge that identifying a target that can rescue the locomotion defect in ADAR-E374A flies may be challenging. Nonetheless, it is still worthwhile discussing attractive targets (edited or spliced) and testing a few of them individually or together.

2) One of the novelties of Deng et al. is showing that the fly ADAR (which is more similar to ADAR2 than ADAR1) suppresses innate immune responses. The authors nicely ruled out the possibility of Dicer triggering an immune response in ADAR defective flies (Figure 6). We suggest that the authors should perform additional experiments to uncover the mechanism of how the fly ADAR suppresses immune responses.

- To address if ADAR plays a role in suppressing the fly homolog of NFκB, the authors crossed ADAR5G1 to Rel double mutants, but they could not obtain any ADAR-5G1/Rel double mutants. Alternatively, the authors can try crossing ADAR-E374A to Rel double mutants. Or they could conditionally knock-down Rel to bypass the lethality. Or could they try depleting one Rel at a time?

- If Rel is conclusively ruled out or if it does not completely down-regulate immune responses in ADAR defective flies, can other genes in the drosophila immune network be involved in inducing immune responses in ADAR-E374A flies? For example, could ADAR suppress DIF or JAK-STAT pathways in these flies? The authors can try deleting components of these pathways in the ADAR-E374A fly.

3) Is the neurodegeneration observed in ADAR deficient flies (Figure 3) due to cell death, lack of cell proliferation, or lack of cell differentiation?

4) Discussion point: The mechanism of innate immune induction during ADAR depletion may be completely different in flies vs. mammals. The immune responses in flies (which appears to be overall subtle, Figure 4A) may not be triggered by an endogenous RNA, but rather may be a general stress response to disrupted homeostasis in ADAR deficient flies. These possibilities should be discussed. For example, could the immune response in ADAR-5G1 or ADAR-E374A flies be triggered by neurodegeneration (cell death)? Or, as the authors mention an immune response may be causing the neurodegeneration. That being said, to address these important questions, experiments proposed in #1 or #2 may be helpful. Experiments in #1 may generate a system where the locomotion defect/neurodegeneration is suppressed in ADAR defective flies. Experiments in #2, may allow the authors to suppress immune responses in ADAR defective flies. These models, would provide a system to examine if neurodegeneration or the locomotion defect is dependent on the immune response, or

vice versa.

Comments on Figures:

Figure 2:

- Confirm the successful knockdown of ADAR expression when using siRNA.
- Measure hADAR2 protein levels in different groups.
- Instead of in the legend, indicate the statistical significance on the graphs.
- In Figure 2B, is the negative geotaxis assay a behavioral or locomotive test?

Figure 3:

- Quantify the figures and assess significance by statistics.
(Authors mention that the neurodegeneration in ADAR-E374A flies is less severe compared to that in ADAR-5G1 flies. Please quantify this to give readers a better idea of the severity of neurodegeneration in different fly strains)
- Examine how exogenous ADAR protein expression in ADAR5G1;ChAT > ADARE374A flies compare to the endogenous ADAR levels in WT flies.
- Arrows in the images would help to direct the readers.

Figure 6:

- Confirm that Dicer is depleted.
- An explanation/hypothesis in the text as to why Dicer depletion in null mutants (but not editing mutants) increases immune responses would be helpful.

We thank all the reviewers for their insightful comments and constructive suggestions. Please see below for our detailed responses to each suggestion

Reviewer #1 (Remarks to the Author):

This study focuses on ADAR RNA editing in *Drosophila melanogaster*. While flies null for the single *Drosophila* ADAR (dADAR) have been extensively characterized, at least according to this paper, flies expressing a genomic version of a catalytically dead dADAR have not been tested. Crispr technology has allowed the authors to isolate such a mutant, and for the most part the data indicate that RNA editing is the key function of dADAR, as measured by careful behavioral assays and visualization of head sections.

The authors also perform RNAseq experiments to determine genes that are misexpressed in the null and catalytically-dead mutants, and find, interestingly, that a large number of innate immune-related genes are misexpressed. This is interesting because in other organisms ADARs are known to keep cellular dsRNA from triggering an immune response. In mammals the ADAR-related aberrant innate immune response can be rescued by knocking out downstream effectors such as MDA5 or MAVS, or in *C. elegans*, DRH-1. To perform similar experiments in flies, the authors attempt to rescue the misexpression of innate immunity genes with a mutation in Dicer-2, the Dicer known to have antiviral functions in *Drosophila*.

For the most part the experiments are well-performed and represent a significant advance to the field. That said, results with the catalytically dead ADAR mutant are not so surprising, and the most interesting results are those showing aberrant expression of innate immune genes. As detailed below, the quality of the paper would increase if the authors expanded the latter section, at the least providing additional discussion.

We thank reviewer #1 for the insightful and helpful comments. Please see below for our replies to the suggestions and comments.

Specific comments:

1. The authors find that the behavioral defects in the catalytically dead AdarE374A mutant are only partially rescued by Adar $\frac{3}{4}$ S, and propose that this may be due to a dominant-negative effect of the catalytic mutant. To enable an RNAi knockdown that is specific for the catalytically dead dADAR2, the authors rescue with human ADAR2, and then knockdown the dAdarE374A sequence. A very slight rescue in the locomotion assay (Fig. 2A) is observed. Because the rescue is very slight, one wonders if this rescue is significant. The authors should report some measure of statistical significance for Fig. 2A (I think the p-values that appear at the end of the legend refer to panel B, but regardless, the relevant one is missing). As they authors note, the small increase could also be explained by the fact that the RNAi knockdown is only partial. To lend confidence in the significance of the rescue the authors could directly evaluate the change in mRNA levels of AdarE374A. Unless confidence in the observation is increased, the authors should not say that the data "... gives clear evidence...".

We have measured the mRNA levels of AdarE374A by qRT-PCR and level of ADAR2 by immunoblot and also added the Statistics in Figure 2A. The difference is statistically significant. Having two or more UAS drivers within one fly strain causes problems with expression levels. This has been observed with other researchers and we now mention this in the manuscript.

2. The observation that innate immune genes are misexpressed in dADAR mutants is the most

interesting part of the paper, and the validation of these expression changes using qRT-PCR in whole flies is very nice. That said, the quality of the paper would increase if the authors could expand this part of the paper by adding additional experiments that might increase mechanistic insight, or at the least, additional discussion. I am particularly focused on results with the Dicer-2 mutant. As the authors certainly know, *Drosophila* Dicer-2 processes endogenous siRNAs as well as viral siRNAs, and the requirements for each are slightly different. Is the mutant the authors used known to affect a Dicer2-mediated antiviral response? For example, does this mutant lead to a decrease in viral siRNAs or viral load? If the Dicer2(R416X) mutant only affected production of endogenous siRNAs, it could actually lead to a greater immune response (as the authors observe) since it would free up Dicer-2 to act on viral dsRNA.

The mutant which we used for Dicer-2 was created by EMS mutagenesis and has a point mutation expected to result in a truncated protein which has a part of N-terminal helicase domain. It is known that this mutant is defective for endogenous siRNA production and also this mutant is susceptible to viral infection, consistent with the loss of RNaseIII domains and later parts of Dicer-2. Dcr-2^{R416X} flies have a very high viral titer when treated with virus.

Vago is an antiviral protein whose transcript is activated by Dicer-2. Deddouche *et al.* viewed the Dcr-2^{R416X} mutant as a Dcr-2 null mutant; they found that his mutant partly but not completely prevents Vago induction. Since this mutant and also Dcr-2A500V affect the helicase domain they concluded that the helicase domain of Dicer-2 is required for Vago induction but some other sensor may also contribute. The data is also consistent with our own interpretation that Dcr-2 is the key sensor but even these truncated or mutated helicase domains are sufficient for the sensor function.

We agree with the reviewer that the greater immune response we see when using the Adar, Dcr-2^{R416X} flies is because these flies are defective in siRNA production leaving the Dcr-2 helicase domain free to sense the dsRNA in the fly which leads to greater immune response.

We have discussed this above point in the manuscript. We now have included data showing the upregulation of innate immune transcripts in *Drosophila* heads in *Adar* mutants which is significantly stronger than the whole body.

In many respects the connection of *Drosophila* ADAR to the immune response may be more similar to other invertebrates than mammals. Indeed, in *C. elegans* the loss of ADARs increases the number of siRNAs (even in the absence of viral infection), and the resulting increase in silencing can be rescued with a second mutation in RNAi factors that act with Dicer in the antiviral response. Is it known that loss of dADAR increases levels of siRNAs and that this can be rescued by the Dicer2^{R416X}? Finally, it would be helpful if the authors provided a bit more discussion of the particular innate-immune genes observed to have altered expression in the dADAR mutants. Was the vago gene that has previously been connected to altered immune response in *Drosophila* misexpressed?

It seems very likely that in the *Adar* mutant there is an increase in the number of siRNAs as dAdar expression reduces the effectiveness of *white* gene silencing in the eye when a *white* dsRNA is expressed there (Heale *et al.* EMBO J). Human ADAR1 does this even more strongly and has been proposed to interact directly with the helicase domain of human Dicer.

We are very thankful to the reviewer for suggesting that we check the expression of the *Vago* transcript. We found that *Vago* was somewhat upregulated in our sequencing data but when we analyzed the expression of *Vago* by RT-PCR we found huge upregulation in the *Adar* null mutant compared to the wild type; *Vago* was also upregulated in the *Adar*^{EA} mutant but less compared to the null.

Since we saw that *Vago* was upregulated and it gets activated by Dicer-2, we knocked down Dicer-2 in the head and saw that it was able to rescue the immune gene phenotype and reduce aberrant activation of AMPs and *Vago*.

The above result gave us a potential hypothesis that the unedited dsRNA which are present in the null mutant are recognized by the Dcr-2 which is sensing it by its and DExD/H-box helicase domain and activating *Vago* protein.

Minor comments:

1. On line 224 I think the reference should be to Fig. 5A, left.
2. On line 241 I think the Dicer2 allele should be 416X
3. The fonts in Figure 2 are very hard to read.

We have made these modifications.

Reviewer #2 (Remarks to the Author):

ADAR RNA editing enzyme binds double stranded RNA for deaminating adenosines to inosine. Inosine is decoded as guanosine which can alter the amino acid sequence of the encoded protein, but in double stranded RNA inosine also disrupts base-pairing which tunes down the immune response against double stranded RNA in mice.

Here, Deng et al now determine that ADAR in *Drosophila* requires its editing capacity to exert its function as mutants for catalytically inactive ADAR show neurodegenerative and locomotion phenotypes similar to ADAR null mutants. When this catalytically inactive ADAR protein is overexpressed, however, the ADAR RNA binding function can suppress these phenotypes indicating editing independent functions of ADAR.

In addition, immune genes are induced in ADAR catalytic dead and null mutants. This phenotype is reminiscent of the induction of immune genes in mice ADAR1 mutants, where the double-stranded RNA sensor MDA5 is required to turn on immune genes. The closest homolog of MDA5 in flies, DICER2, however, is not responsible for turning on immune genes. Hence, flies use a different pathway to turn on immune genes and this function relies on ADAR editing.

The work by Deng et al addresses the important question of whether ADAR exerts functions by binding double-stranded RNA that are independent of RNA editing. To address this question they use a *Drosophila* model, which has only one ADAR, unlike mice, which have three. These experiments are well performed and they obtained a clear cut answer that editing is required for ADAR function.

These data nicely complement previous work done in the more complex situation of mice with having two active ADARs and highlight the importance of RNA editing.

Importantly, the work of Deng et al further discovers an evolutionary conserved function of ADAR in the regulation of immune genes. Immune genes are also turned on in *Drosophila*, but unlike in vertebrates this seems not to be mediated by DICER2, the sensor of double-stranded RNA in flies.

We would like to thank the reviewer for carefully reviewing our Manuscript and giving us a positive feedback, we have carefully gone through the manuscript and added the data and corrected the issues.

Minor issues

P1, line 61: should mention that lethality can be rescued by the edited cDNA

We have done this.

.....

Fig 6B: In the double mutant of Adar null, DICER2, the tested immune genes are dramatically up-regulated, but this induction is not present in the ADAR E374A DICER2 mutant. Does the RNA binding capacity of ADAR E374A prevent this induction? To answer this question the DICER2 mutant control needs to be added. Also, a reference for the allele used is missing.

We have done this

The discussion contains an number of results. These should be moved to the results section.

We have done this.

Reviewer #3 (Remarks to the Author):

ADAR null model organisms have uncovered many important functions of ADAR. However, it was not clear whether these functions of ADAR were editing dependent or independent. Previously, a transgenic mouse defective in ADAR editing (ADAR-E861A) was shown to be embryonic lethal, demonstrating that the ADAR editing activity plays an essential role in mammals (Liddicoat et al. Science 2015). To examine the function of ADAR editing in flies, Deng et al. generated a novel drosophila strain carrying a point mutation in the catalytic domain of ADAR (ADAR-E374A). With ADAR-E374A flies, Deng et al. demonstrated that ADAR editing is entirely responsible for preventing locomotion defects in flies and mostly (but not entirely) responsible for preventing neurodegeneration. Furthermore, they made an unexpected finding that ADAR editing plays a critical role in suppressing immune responses in flies. This is surprising since drosophila ADAR is similar to mammalian ADAR2 whose known roles lie in the brain, and is less similar to mammalian ADAR1, which plays a role in suppressing innate immune responses. Taken together, this study reveals that in drosophila, ADAR's RNA editing activity is required for proper brain and immune function. The ADAR-E374A drosophila model system is novel, and we believe that the authors' findings are interesting. However, mechanisms underlying the locomotion defect or induction of immunity in ADAR deficient flies should be further explored in depth. Furthermore, the quality of data (quantifying images, confirming protein expression, confirming successful knock-down with siRNA etc.) in some figures need to be significantly improved. Please see our comments below.

Major comments:

1) Similar to how the ADAR2^{-/-} mice defects were rescued by introducing the edited version of Gria2 (Higuchi et al. Nature 2000), could the locomotion defect in ADAR-E374A flies be rescued (at least partially if not entirely) by expressing an edited gene? Are there any edited neuronal genes that are strong candidates? Alternatively, since the authors mention that differential exon usage was most

significant among neuronal genes (Supplementary Figure 3), could expressing a properly spliced transcript rescue the locomotion defect in ADAR-E374A flies? We acknowledge that identifying a target that can rescue the locomotion defect in ADAR-E374A flies may be challenging. Nonetheless, it is still worthwhile discussing attractive targets (edited or spliced) and testing a few of them individually or together.

As we mention in the introduction the rescue of *Adar2* by pre-edited *Gria2* at the Q/R site is an exception and has never been observed for any other transcript in any organism or tissue for 30 years. For approximately 10 years researchers looked for a similar critical editing event to rescue *Adar1* lethality. We were the first to take a genetic approach and demonstrate that for *Adar1* it is highly unlikely that one critical edited transcript exists.

There are approximately 1000 specific editing events in approximately 600 transcripts in *Drosophila*. Many of these are expressed in the nervous system and some are candidate for the lack of locomotion. However, there are often up to 10 or more editing events per transcript, so we would have to knock-in each of these individual editing events in multiple transcripts. This would take years and may not even be successful. For this reason, the lack of locomotion is not the focus of this paper.

2) One of the novelties of Deng et al. is showing that the fly ADAR (which is more similar to ADAR2 than ADAR1) suppresses innate immune responses. The authors nicely ruled out the possibility of Dicer triggering an immune response in ADAR defective flies (Figure 6). We suggest that the authors should perform additional experiments to uncover the mechanism of how the fly ADAR suppresses immune responses.

- To address if ADAR plays a role in suppressing the fly homolog of NFkB, the authors crossed ADAR5G1 to Rel double mutants, but they could not obtain any ADAR-5G1/Rel double mutants. Alternatively, the authors can try crossing ADAR-E374A to Rel double mutants. Or they could conditionally knock-down Rel to bypass the lethality. Or could they try depleting one Rel at a time?

- If Rel is conclusively ruled out or if it does not completely down-regulate immune responses in ADAR defective flies, can other genes in the drosophila immune network be involved in inducing immune responses in ADAR-E374A flies? For example, could ADAR suppress DIF or JAK-STAT pathways in these flies? The authors can try deleting components of these pathways in the ADAR-E374A fly.

We found out that *Vago* gene which is antiviral in nature and it is activated by Dicer-2 was highly upregulated in *Adar* mutant. Deddouche *et al.* found R416X Mutant still has a part of DExD/H-box helicase domain and DExD/H-box helicase domain of Dicer-2 was required for *Vago* induction. So, we used a mutant which still has the ability to sense the dsRNA and activate the antiviral response. Since we saw that *Vago* was upregulated and it gets activated by Dicer-2, we knocked down Dicer-2 in the brain and saw that it was able to rescue the immune gene phenotype. The reviewer asked us to check if knockdown of *Rel* and *Dif* which are activated by IMD and Toll pathway respectively can rescue the immune gene phenotype in ADAR-E374A flies. We found that knocking down *Rel* and *Dif* doesn't rescue the immune gene phenotype, whereas knocking down STAT does rescue the immune gene phenotype.

3) Is the neurodegeneration observed in ADAR deficient flies (Figure 3) due to cell death, lack of cell proliferation, or lack of cell differentiation?

We have another manuscript that is currently in review, in which we have shown that the neurodegeneration in the *Adar* mutant can be rescued by the increasing autophagy and not by apoptosis. We have uploaded this manuscript for additional information for the reviewer.

4) Discussion point: The mechanism of innate immune induction during ADAR depletion may be completely different in flies vs. mammals. The immune responses in flies (which appears to be overall subtle, Figure 4A) may not be triggered by an endogenous RNA, but rather may be a general stress response to disrupted homeostasis in ADAR deficient flies. These possibilities should be discussed. For example, could the immune response in ADAR-5G1 or ADAR-E374A flies be triggered by neurodegeneration (cell death)? Or, as the authors mention an immune response may be causing the neurodegeneration. That being said, to address these important questions, experiments proposed in #1 or #2 may be helpful. Experiments in #1 may generate a system where the locomotion defect/neurodegeneration is suppressed in ADAR defective flies. Experiments in #2, may allow the authors to suppress immune responses in ADAR defective flies. These models, would provide a system to examine if neurodegeneration or the locomotion defect is dependent on the immune response, or vice versa.

Thank you for this comment. The Discussion now covers this in more detail. In addition, many of these points are dealt with in our other manuscript which is on *Adar* and endosomal-autophagy. We were unsuccessful in our attempt to submit both of these manuscripts together as we believe together they make a more complete story

Comments on Figures:

Figure 2:

- Confirm the successful knockdown of ADAR expression when using siRNA.

We have done this

- Measure hADAR2 protein levels in different groups.

We have done this

- Instead of in the legend, indicate the statistical significance on the graphs.

We have corrected this

- In Figure 2B, is the negative geotaxis assay a behavioral or locomotive test?

It is a locomotive test

Figure 3:

- Quantify the figures and assess significance by statistics.

(Authors mention that the neurodegeneration in ADAR-E374A flies is less severe compared to that in ADAR-5G1 flies. Please quantify this to give readers a better idea of the severity of neurodegeneration in different fly strains)

We do not discuss this in the revised manuscript. We do not have a good method to quantitate this as there is too much variation in the size of the vacuoles. Fig. 3 simply shows that *Adar*^{E374A} still has neurodegeneration.

- Examine how exogenous ADAR protein expression in ADAR5G1;ChAT > ADARE374A flies compare to the endogenous ADAR levels in WT flies.

We have checked the expression of *Adar* by RT-PCR.

- Arrows in the images would help to direct the readers.

We have done this

Figure 6:

- Confirm that Dicer is depleted.

We have done this

- An explanation/hypothesis in the text as to why Dicer depletion in null mutants (but not editing mutants) increases immune responses would be helpful.

When we overexpressed the *Adar*^{EA}, it is able to rescue the immune gene phenotype, this gives us an indication that there is an editing independent function of *Adar*. Therefore when *Adar*^{EA} protein is present in the flies it is able to bind to its target transcript so there is less unedited RNA present in the *ADAR*^{E374A} flies compared to the null mutant.

REVIEWERS' COMMENTS:

Reviewer #1 (Remarks to the Author):

The authors have carefully addressed prior criticisms, and the quality of the paper has increased. The idea that Dicer's helicase domain is involved in inducing expression of innate immunity genes in *Drosophila* is exciting, albeit future studies will be required to conclusively show this. In this regard, the authors might want to tone down some of their statements about the role of Dicer's helicase domain in innate immunity. For example, since we don't yet know the levels of Dicer protein in the Dcr-2R416X mutant, and whether dsRNA binds this protein, the authors should tone down the sentence below which appears on page 12:

"We conclude that Dcr-2 is the antiviral innate immune sensor aberrantly activated by unedited dsRNA in Adar mutants and that the dsRNA-activated helicase domain of Dcr-2 likely mediates this activation, possibly by interaction with other proteins."

Similarly, the sentence below, also on page 12, would be more supported if the authors had additional experiments monitoring levels of vago after RNAi to Dicer, or in the AdarE374A; Dcr-2R416X mutant strain. Certainly if the authors have done these experiments, they should add them.

"We conclude that aberrant innate immune induction in Adar mutants is likely to involve innate immune signaling from Dcr-2 to activate expression of Vago which acts as an interferon-like secreted antiviral signaling protein activating the Jak-STAT pathway 37,41."

Finally, it would be helpful for the reader if the authors clarified the levels of the various transcripts in the w1118 strain for Supplementary Fig. 4. The legend says that the levels were "compared" to the levels in w1118, but it is unclear what this means. Does "50" on the y-axis mean 50 times the wild type level? It is unclear if the wt transcript levels were the same as various transcripts shown in a, or much less. This seems very important since if the wt levels were similar to all transcripts except in the AdarE374A, AdarE374A; ChAT>Stat93E RNAi samples, there is the possibility that they are looking at an ADAR independent effect.

Brenda Bass

Reviewer #2 (Remarks to the Author):

The authors have adequately addressed my comments and those of the other reviewers and this exciting MS can be published.

Since the authors have a second complementary manuscript addressing the neurodegeneration phenotype in ADAR mutants in detail, they should upload both manuscripts to BioRxiv and cross-

reference them.

Reviewer #3 (Remarks to the Author):

In this revised manuscript, Deng et al. has addressed our comments adequately. This study uncovers how ADAR's editing activity is critical to prevent neurodegeneration and aberrant innate immune responses in flies. Previously, we inquired about the mechanism of how ADAR suppresses innate immune responses. In this revised manuscript, Deng et al. demonstrates that ADAR's suppression of Dicer2 is required to suppress innate immune responses in flies. Furthermore, the overall quality of the data in the manuscript has also significantly improved. We are satisfied with the revised manuscript.

We only have minor comments/suggestions on some figures.

- Supplementary Figure 4B: Please show successful depletion of Dicer-2 in flies expressing Dcr-2 RNAi.
- In Figure 6C I think it's better to indicate statistical significance between the,
 - a) the blue bar (ADAR5G1) and orange bar (ADAR5G1;ChAT > Dcr-2 RNAi)
 - b) the gray bar (ADARE374A) vs. green bar (ADARE374A;ChAT > Dcr-2 RNAi)

REVIEWERS' COMMENTS:

We thank all the reviewers for their insightful comments and constructive suggestions that has help us to increase the quality of the paper.

Reviewer #1 (Remarks to the Author):

The authors have carefully addressed prior criticisms, and the quality of the paper has increased. The idea that Dicer's helicase domain is involved in inducing expression of innate immunity genes in *Drosophila* is exciting, albeit future studies will be required to conclusively show this. In this regard, the authors might want to tone down some of their statements about the role of Dicer's helicase domain in innate immunity. For example, since we don't yet know the levels of Dicer protein in the Dcr-2R416X mutant, and whether dsRNA binds this protein, the authors should tone down the sentence below which appears on page 12:

"We conclude that Dcr-2 is the antiviral innate immune sensor aberrantly activated by unedited dsRNA in Adar mutants and that the dsRNA-activated helicase domain of Dcr-2 likely mediates this activation, possibly by interaction with other proteins."

Similarly, the sentence below, also on page 12, would be more supported if the authors had additional experiments monitoring levels of *vago* after RNAi to Dicer, or in the AdarE374A; Dcr-2R416X mutant strain. Certainly if the authors have done these experiments, they should add them.

"We conclude that aberrant innate immune induction in Adar mutants is likely to involve innate immune signaling from Dcr-2 to activate expression of *Vago* which acts as an interferon-like secreted antiviral signaling protein activating the Jak-STAT pathway 37,41."

We agree with the reviewer and we have modified and toned down the sentences mentioned above. And we have added the RT-PCR results result monitoring levels of *Dicer-2* and *Vago* transcripts after *Dicer-2* RNAi in Supplementary Figure 4b.

Finally, it would be helpful for the reader if the authors clarified the levels of the various transcripts in the w1118 strain for Supplementary Fig. 4. The legend says that the levels were "compared" to the levels in w1118, but it is unclear what this means. Does "50" on the y-axis mean 50 times the wild type level? It is unclear if the wt transcript levels were the same as various transcripts shown in a, or much less. This seems very important since if the wt levels were similar to all transcripts except in the AdarE374A, AdarE374A; ChAT>Stat93E RNAi samples, there is the possibility that they are looking at an ADAR independent effect.

We have modified the figure which is now Supplementary Figure 5 and added the data for wild type, this would make the figure much clearer.

Brenda Bass

Reviewer #2 (Remarks to the Author):

The authors have adequately addressed my comments and those of the other reviewers and this exciting MS can be published.

Since the authors have a second complementary manuscript addressing the neurodegeneration phenotype in ADAR mutants in detail, they should uploaded both manuscripts to BioRxiv and cross-reference them.

We would like to thank the reviewer for carefully reviewing our manuscript and giving us positive feedback. The second manuscript is accepted at BMC Biology and should be online soon therefore we don't think its necessary to upload the manuscript to BioRxiv. We now comment on the second manuscript in the Discussion.

Reviewer #3 (Remarks to the Author):

In this revised manuscript, Deng et al. has addressed our comments adequately. This study uncovers how ADAR's editing activity is critical to prevent neurodegeneration and aberrant innate immune responses in flies. Previously, we inquired about the mechanism of how ADAR suppresses innate immune responses. In this revised manuscript, Deng et al. demonstrates that ADAR's suppression of Dicer2 is required to suppress innate immune responses in flies. Furthermore, the overall quality of the data in the manuscript has also significantly improved. We are satisfied with the revised manuscript.

We thank reviewer for the insightful and helpful comments.

We only have minor comments/suggestions on some figures.

- Supplementary Figure 4B: Please show successful depletion of Dicer-2 in flies expressing Dcr-2 RNAi. We have added the data in Supplementary Figure 4a.

- In Figure 6C I think it's better to indicate statistical significance between the,
a) the blue bar (ADAR5G1) and orange bar (ADAR5G1;ChAT > Dcr-2 RNAi)
b) the gray bar (ADARE374A) vs. green bar (ADARE374A;ChAT > Dcr-2 RNAi)

We have modified the figure and mentioned that the that decrease in the aberrant induction of AMP transcripts in *Adar*^{E374A};ChAT>*Dicer2*^{RNAi} and *Adar*^{5G1};ChAT>*Dicer2*^{RNAi} heads compared to *Adar*^{E374A} and *Adar*^{5G1} respectively is statistically significant in the Result as well as in the Figure legend .